

# 1 Impact of Timanian thrust systems on the late
# 2 Neoproterozoic–Phanerozoic tectonic evolution of the
# 3 Barents Sea and Svalbard

Jean-Baptiste P. Koehl[1,2,3,4], Craig Magee[5], Ingrid M. Anell[6]
[1]Centre for Earth Evolution and Dynamics (CEED), University of Oslo, PO Box 1028 Blindern, N-0315 Oslo,
Norway.
[2]Department of Geosciences, UiT The Arctic University of Norway in Tromsø, N-9037 Tromsø, Norway.
[3]Research Centre for Arctic Petroleum Exploration (ARCEx), UiT The Arctic University of Norway in Tromsø, N-
9037 Tromsø, Norway.
[4]CAGE – Centre for Arctic Gas Hydrate, Environment and Climate, UiT The Arctic University of Norway in
Tromsø, N-9037 Tromsø, Norway.
[5]School of Earth Science and Environment, University of Leeds, Leeds, LS2 9JT, United Kingdom.
[6]Department of Geosciences, University of Oslo, P.O. Box 1047 Blindern, N-0316 Oslo, Norway.
**Correspondence:** Jean-Baptiste P. Koehl (jean-baptiste.koehl@uit.no)
**Abstract**
The Svalbard Archipelago is composed of three basement terranes that record a complex
Neoproterozoic–Phanerozoic tectonic history, including four contractional events (Grenvillian,
Caledonian, Ellesmerian, and Eurekan) and two episodes of collapse- to rift-related extension
(Devonian–Carboniferous and late Cenozoic). These three terranes are thought to have accreted
during the early–mid Paleozoic Caledonian and Ellesmerian orogenies. Yet recent
geochronological analyses show that the northwestern and southwestern terranes of Svalbard both
record an episode of amphibolite (–eclogite) facies metamorphism in the latest Neoproterozoic,
which may relate to the 650–550 Ma Timanian Orogeny identified in northwestern Russia, northern
Norway and the Russian Barents Sea. However, discrete Timanian structures have yet to be
identified in Svalbard and the Norwegian Barents Sea. Through analysis of seismic reflection, and
regional gravimetric and magnetic data, this study demonstrates the presence of continuous, several
kilometers thick, NNE-dipping, deeply buried thrust systems that extend thousands of kilometers
from northwestern Russia to northeastern Norway, the northern Norwegian Barents Sea, and the
Svalbard Archipelago. The consistency in orientation and geometry, and apparent linkage between



these thrust systems and those recognized as part of the Timanian Orogeny in northwestern Russia
and Novaya Zemlya suggests that the mapped structures are likely Timanian. If correct, these
findings would indicate that Svalbard's three basement terranes and the Barents Sea were accreted
onto northern Norway during the Timanian Orogeny and should, hence, be attached to Baltica and
northwestern Russia in future Neoproterozoic–early Paleozoic plate tectonics reconstructions. In
the Phanerozoic, the study suggests that the interpreted Timanian thrust systems represented major
preexisting zones of weakness that were reactivated, folded, and overprinted by (i.e., controlled the
formation of new) brittle faults during later tectonic events. These faults are still active at present
and can be linked to folding and offset of the seafloor.

**Introduction**

Recognizing and linking tectonic events across different terranes is critical to plate
reconstructions. In the latest Neoproterozoic (at ca. 650–550 Ma), portions of northwestern Russia
(e.g., Timan Range and Novaya Zemlya) and the Russian Barents Sea were accreted to northern
Baltica by top-SSW thrusting during the Timanian Orogeny (Olovyanishnikov et al., 2000;
Kostyuchenko et al., 2006). Discrete Timanian structures with characteristic WNW–ESE strikes
are sub-orthogonal to the N–S-trending Caledonian grain formed during the closure of the Iapetus
Ocean (Gee et al., 1994; Witt-Nilsson et al., 1998; Johansson et al., 2004; 2005). Thus far,
Timanian structures have only been identified in onshore–nearshore areas of northwestern Russia
and northeastern Norway and offshore in the Russian Barents Sea and southeasternmost Norwegian
Barents Sea (Barrère et al., 2009, 2011; Marello et al., 2010; Gernigon et al., 2018; Hassaan et al.,
2020a, 2020b). Therefore, the nature of basement rocks in the northern and southwestern
Norwegian Barents Sea remains debatable. Some studies suggest a NE–SW-trending Caledonian
suture within the Barents Sea (Gudlaugsson et al., 1998; Gee and Teben'kov, 2004; Breivik et al.,
2005; Gee et al., 2008; Knudsen et al., 2019), whereas others argue for a swing into a N–S trend
and merging of Norway and Svalbard's Caledonides, which are expected to continue into northern
Greenland (Ziegler, 1988; Gernigon and Brönner, 2012; Gernigon et al., 2014). Regardless, these
models solely relate basement structures in the northern and southwestern Norwegian Barents Sea
to the Caledonian Orogeny, implying that Laurentia and Svalbard were not involved in the
Timanian Orogeny and were separated from Baltica by the Iapetus Ocean in the latest



Neoproterozoic (Torsvik and Trench, 1991; Cawood et al., 2001; Cocks and Torsvik, 2005; Torsvik
et al., 2010; Merdith et al., 2021).
Nonethelesss, geochronological data yielding Timanian ages suggest that deformation and
metamorphism contemporaneous of the Timanian Orogeny affected parts of the Svalbard
Archipelago and Laurentia and, possibly, all Arctic regions (Estrada et al., 2018; Figure 1a): (1)
eclogite facies metamorphism (620–540 Ma; Peucat et al., 1989; Dallmeyer et al., 1990b) and
eclogite facies xenoliths of mafic–intermediate granulite in Quaternary volcanic rocks are found in
northern Spitsbergen (648–556 Ma; Griffin et al., 2012); (2) amphibolite facies metamorphism
(643 ± 9 Ma; Majka et al., 2008, 2012, 2014; Mazur et al., 2009) and WNW–ESE-striking shear
zones like the Vimsodden–Kosibapasset Shear Zone occur in southwestern Spitsbergen (600–537
Ma; Manecki et al., 1998; Faehnrich et al., 2020); and (3) xenoliths of the subduction-related
Midtkap igneous suite in northern Greenland yield Timanian ages (628–570 Ma; Rosa et al., 2016;
Estrada et al., 2018). In addition, several recent studies also show the presence of NW–SE- to E–
W-trending basement grain in the Norwegian Barents Sea, which could possibly represent
Timanian fabrics and structures (Figure 1b; Barrère et al., 2009, 2011; Marello et al., 2010; Klitzke
et al., 2019). Following these developments, a few paleo-plate reconstructions now place Svalbard
together with Baltica in the latest Neoproterozoic–Paleozoic (e.g., Vernikovsky et al., 2011).
To test the origin of basement grain in the northern Norwegian Barents Sea and Svalbard,
the present study focuses on several kilometers deep structures identified on seismic data and
correlated using regional gravimetric and magnetic data. These newly identified structures trend
WNW–ESE, i.e., parallel to the Timanian structural grain in northwestern Russia and northern
Norway (Figure 1a–c). The structures are described and interpreted based on their geometry and
potential kinematic indicators, and are compared to well-known examples of Caledonian and
Timanian fabrics and structures elsewhere, e.g., onshore Norway (e.g., Trollfjorden–Komagelva
Fault Zone; Siedlecka and Siedlecki, 1967; Siedlecka, 1975), in Svalbard (e.g., Atomfjella
Antiform; Witt-Nilsson et al., 1998), in northwestern Russia (Central Timan Fault; Siedlecka and
Roberts, 1995; Olovyanishnikov et al., 2000; Kostyuchenko et al., 2006), and in the southern
Norwegian Barents Sea (Barrère et al., 2011; Gernigon et al., 2014) and the Russian Barents Sea
(Baidaratsky fault zone; Lopatin et al., 2001; Korago et al., 2004). A scenario involving several
episodes of deformation (Timanian Orogeny, and reactivation and overprinting during the
Caledonian Orogeny, Devonian–Carboniferous extension, Triassic extension, Eurekan tectonism,



and present-day tectonism) is proposed and the implications for the tectonic evolution of the
Barents Sea and the Svalbard Archipelago and associated basins (e.g., Ora and Olga basins; Anell
et al., 2016) are discussed.
Should our interpretation of discrete Timanian structures throughout the Norwegian
Barents Sea and Svalbard be validated, it would support accretion of these terranes to Baltica in
the late Neoproterozoic and place the Caledonian suture farther west than is commonly suggested
(e.g., Breivik et al., 2005; Gernigon et al., 2014), thus leading to a major revision of plate tectonics
models. In addition, constraining the extent and reactivation history of such faults may shed some
light on their influence on younger tectonic events, such as Caledonian, Ellesmerian and Eurekan
contraction, Devonian–Carboniferous collapse–rifting, and late Cenozoic breakup and ongoing
extension.

**Geological setting**
***Timanian Orogeny***
The Timanian Orogeny corresponds to a ca. 650–550 Ma episode of NNE–SSW
contractional deformation that affected northwestern Russia and northeastern Norway. During this
tectonic episode, crustal-scale, WNW–ESE-striking, NNE-dipping thrusts systems with south-
southwestwards transport direction (top-SSW; Siedlecka and Siedlecki, 1967; Siedlecka, 1975;
Figure 1b), accreted portions of the Russian Barents Sea and northwestern Russia onto northeastern
Baltica, including Novaya Zemlya, Severnaya Zemlya, the Kanin Peninsula, the Timan Range, and
the Kola Peninsula (Siedlecka and Roberts, 1995; Olovyanishshnikov et al., 2000; Roberts and
Siedlecka, 2002; Gee and Pease, 2004; Kostyuchenko et al., 2006; Lorenz et al., 2008; Marello et
al., 2013) and the Varanger Peninsula in northeastern Norway (Siedlecka and Siedlecki, 1967;
Siedlecka, 1975; Roberts and Olovyanishshnikov, 2004; Herrevold et al., 2009; Drachev, 2016;
Figure 1a). Major Timanian thrusts include the Baidaratsky fault zone in the Russian Barents Sea
and Novaya Zemlya (Figure 1a–b; Eldholm and Ewing, 1971, their figure 4 profile C–D; Lopatin
et al., 2001; Korago et al., 2004; Drachev, 2016), the Central Timan Fault on the Kanin Peninsula
and the Timan Range (Siedlecka and Roberts, 1995; Olovyanishnikov et al., 2000; Kostyuchenko
et al., 2006), and the Trollfjorden–Komagelva Fault Zone in northern Norway (Siedlecka and
Siedlecki, 1967; Siedlecka, 1975; Herrevold et al., 2009).



*Accretion of Svalbard basement terranes in the early Paleozoic*

The Svalbard Archipelago consists of three Precambrian basement terranes, some of which

show affinities with Greenland (northwestern and northeastern terranes), whereas others are
possibly derived from Pearya (southwestern terrane; Harland and Wright, 1979; Ohta et al., 1989;
Gee and Teben'kov, 2004; Labrousse et al., 2004; Piepjohn et al., 2013; Fortey and Bruton, 2013).
These terranes are inferred to have accreted during the mid-Paleozoic Caledonian (collision of
Greenland with Svalbard and Norway at ca. 460–410 Ma; Horsfield, 1972; Dallmeyer et al., 1990a;
Johansson et al., 2004, 2005; Faehnrich et al., 2020) and Late Devonian Ellesmerian orogenies
(Piepjohn, 2000; Majka and Kosminska, 2017). In these models, accretion was facilitated via
hundreds of kilometers of displacement along (arcuate) strike-slip faults, such as the Billefjorden
Fault Zone (Harland, 1969; Harland et al., 1992; Labrousse et al., 2008) and the Lomfjorden Fault
Zone (Piepjohn et al., 2019; Figure 2), although other studies suggest more limited strike-slip
displacement (Lamar et al., 1986; Manby and Lyberis, 1992; Manby et al., 1994; Lamar and
Douglass, 1995). These large (strike-slip?) faults are assumed to have extended thousands of
kilometers southwards and to represent the continuation of Caledonian faults in Scotland (Norton
et al., 1987; Dewey and Strachan, 2003). Caledonian contraction resulted in the formation of large
fold and thrust complexes, such as the Atomfjella Antiform in northeastern Spitsbergen (Gee et al.,
1994; Witt-Nilsson et al., 1998) and the Rijpdalen Anticline in Nordaustlandet (Johansson et al.,
2004; 2005; Dumais and Brönner, 2020), whereas Ellesmerian tectonism is thought to have formed
narrow fold and thrust belts, like the Dickson Land and Germaniahalvøya fold-thrust zones
(McCann, 2000; Piepjohn, 2000; Dallmann and Piepjohn, 2020).

In northern Norway, Timanian thrusts were reactivated–overprinted in subsequent tectonic

events (e.g., Caledonian Orogeny and late–post-Caledonian collapse–rifting) as dominantly strike-
to oblique-slip faults (Siedlecka and Siedlecki, 1971; Roberts et al., 1991; Herrevold et al., 2009;
Rice, 2014). A notable example is the folding and reactivation of Timanian fabrics and structures
(e.g., Trollfjorden–Komagelva Fault Zone) during the Caledonian Orogeny (Siedlecka and
Siedlecki, 1971; Herrevold et al., 2009) and intrusion of Mississippian dolerite dykes along steeply
dipping WNW–ESE-striking brittle faults that overprint the Trollfjorden–Komagelva Fault Zone
onshore–nearshore northern Norway (Roberts et al., 1991; Lippard and Prestvik, 1997; Nasuti et
al., 2015; Koehl et al., 2019).





### *Late Paleozoic post-Caledonian collapse and rifting*

In the latest Silurian–Devonian, extensional collapse of the Caledonides led to the
deposition of several kilometers thick sedimentary basins such as the Devonian Graben in northern
Spitsbergen (Gee and Moody-Stuart, 1966; Friend et al., 1966; Friend and Moody-Stuart, 1972;
Murascov and Mokin, 1979; Manby and Lyberis, 1992; Manby et al., 1994; Friend et al., 1997;
McCann, 2000; Dallmann and Piepjohn, 2020). In places, N–S-trending basement ridges
potentially exhumed as metamorphic core complexes along bowed, reactivated detachments, such
as the Keisarhjelmen Detachment in northwestern Spitsbergen (Braathen et al., 2018).
In the latest Devonian–Mississippian, coal-rich sedimentary strata of the Billefjorden
Group were deposited within normal fault-bounded basins throughout Spitsbergen (Cutbill and
Challinor, 1965; Harland et al., 1974; Cutbill et al., 1976; Aakvik, 1981; Koehl and Muñoz-Barrera,
2018; Koehl, 2020a) and the Norwegian Barents Sea (Koehl et al., 2018a; Tonstad, 2018). As rift-
related normal faulting evolved, Pennsylvanian sedimentation was localized into a few, several
kilometers deep, N–S-trending basins like the Billefjorden Trough (Cutbill and Challinor, 1965;
Braathen et al., 2011; Koehl et al., 2021 in review) and the Ora Basin (Anell et al., 2016). In the
Permian, rift-related faulting stopped and platform carbonates were deposited throughout Svalbard
(Cutbill and Challinor, 1965) and the Barents Sea (Larssen et al., 2005).
Overall, the several kilometers thick, late Paleozoic sedimentary succession deposited
during late–post-Caledonian extension buried Proterozoic basement rocks. As a result, these rocks
are sparsely exposed and, thus, difficult to study.

### *Mesozoic sedimentation and magmatism*

In the Mesozoic, Svalbard and the Barents Sea remained tectonically quiet and were only
affected by minor Triassic normal faulting (e.g., Anell et al., 2013; Osmundsen et al., 2014; Ogata
et al., 2018; Smyrak-Sikora et al., 2020). In the Early Cretaceous, Svalbard was affected by a
regional episode of magmatism recorded by the intrusion of numerous dykes and sills of the
Diabasodden Suite (Senger et al., 2013).

### *Early Cenozoic Eurekan tectonism*

The opening of the Labrador Sea and Baffin Bay between Greenland and Arctic Canada in
the early Cenozoic (Chalmers and Pulvercraft, 2001; Oakey and Chalmers, 2012) led to the





collision of northern Greenland with Svalbard and the formation of a fold-and-thrust belt with top-
east thrusts and east-verging folds in western Spitsbergen (Dallmann et al., 1993). In eastern
Spitsbergen, this deformation event is characterized by dominantly thin-skinned deformation
structures, including décollements, some of which showing westwards transport directions
(Andresen et al., 1992; Haremo and Andresen, 1992). Notably, the N–S-striking Agardhbukta
Fault, a major splay/segment of the Lomfjorden Fault Zone, accommodated reverse and, possibly,
strike-slip movements during this event (Piepjohn et al., 2019).

*Late Cenozoic opening of the Fram Strait*
After the end of extension in the Labrador Sea and Baffin Bay, the Fram Strait started to
open in the earliest Oligocene (Engen et al., 2008). Tectonic extension and break-up in the Fram
Strait resulted in the formation of two major, NW–SE-striking transform faults (Lowell, 1972;
Thiede et al., 1990; Figure 1b).

**Methods and datasets**
Seismic surveys from the DISKOS database (see Figure 1b–c and supplement S1 for
location) were used to interpret basement-seated structures and related, younger, brittle overprints
(Figure 3a–f and Figure 4a–h and supplement S2). Other features of interest include potential
dykes, which commonly appear as high positive reflections on seismic data. The geology
interpreted from onshore seismic data was directly correlated to geological maps of the Norwegian
Polar Institute (e.g., Dallmann, 2015). Where possible, interpretation of offshore seismic data was
tied to onshore geological maps and to exploration wells Raddedalen-1 and Plurdalen-1 on
Edgeøya (Bro and Shvarts, 1983; Harland and Kelly, 1997) and to the Hopen-2 well on Hopen
(Anell et al., 2014; Figure 1c and supplement S3). The Raddedalen-1 well penetrated 2823 meters
of Upper Permian to Mississippian or Ordovician strata, the Plurdalen well 2351 meters of Middle
Triassic to (pre-?) Devonian strata, and the Hopen-2 well 2840 meters of Middle–Upper Triassic
to Pennsylvanian strata (Bro and Shvarts, 1983; Harland and Kelly, 1997; Anell et al., 2014; Senger
et al., 2019). Note that the interpretation of lower Paleozoic (Ordovician–Silurian) rocks in the
Raddedalen-1 well by Bro and Shvarts (1983) is preferred to that of upper Paleozoic (Upper
Devonian–Mississippian) by Cambridge Svalbard Exploration (see contrasting interpretations in
Harland and Kelly, 1997). This is based on the more detailed lithological, palynological and





paleontological analyses by the former, and on the strong contrast of the lithologies described in
the well with Devonian–Mississippian successions on Svalbard (Cutbill and Challinor, 1965;
Cutbill et al., 1976; Friend et al., 1997; Dallmann and Piepjohn, 2020).

Only a few examples of seismic sections are included in the present contribution. However,

more interpreted and uninterpreted seismic data are available as supplements (supplements S1–2).
None of the seismic sections were depth-converted, and the thickness are therefore discussed in
seconds (Two-Way Time; TWT). However, local time conversion was performed to tie seismic
wells onshore Edgeøya to seismic section in Storfjorden and depth conversion was performed
locally to evaluate fault displacement. Velocities of Gernigon et al. (2018) were used in these
conversions. Details related to these conversions are shown in supplement S3.

The correlation of kilometer-thick structures discussed in the present contribution was also

tested using gravimetric and magnetic data in cross section (Figure 3a–f) and regional magnetic
and gravimetric data in the northern Norwegian Barents Sea and Svalbard (Figure 5 and supplement
S4) from the Federal Institute for Geosciences and Natural Resources in Germany in map view
(Klitzke et al., 2019). Regional gravimetric and magnetic data are also used to interpret deep
basement fabrics and structures, e.g., regional folds (gravimetric highs commonly associated with
major anticlines of thickened dense basement (i.e., Precambrian) rocks and gravimetric lows with
synclines with less dense sedimentary basins) and large faults that commonly correlate with
elongated gravimetric and/or magnetic anomalies (e.g., Koehl et al., 2019), and to discuss the
relationship of the described structures with known structural trends in onshore basement rocks in
Russia, Norway and Svalbard.

**Results and interpretations**

First, the interpretation of seismic data are described by area, including (1) Storfjorden

(between Edgeøya and Spitsbergen) and the northeastern part of the Norwegian Barents Sea (east
of Edgeøya), (2) Nordmannsfonna to Sassenfjorden onshore–nearshore the eastern–central part of
Spitsbergen, and (3) the northwestern part of the Norwegian Barents Sea between Bjørnøya and
Spitsbergen (Figure 1b–c). Description in each area starts with deep Precambrian basement rocks
and shallow sedimentary rock units, and ends with deep brittle–ductile structures and with shallow
brittle faults. Then, potential field data and regional gravimetric and magnetic anomalies in the
Barents Sea and Svalbard are described, and compared and correlated to seismic data and to major



Timanian and Caledonian fabrics and structures onshore northwestern Russia, Svalbard and
Norway. Please see high resolution versions of all the figures and supplements on DataverseNO
(doi.org/10.18710/CE8RQH).

***Structures in the northwestern–northeastern Norwegian Barents Sea, Storfjorden and central–***
***eastern Spitsbergen***
*Storfjorden and northeastern Norwegian Barents Sea*
Folded Precambrian–lower Paleozoic basement rocks

Seismic facies at depths of 2–6 seconds (TWT) typically comprise successions of laterally

discontinuous (< three kilometers long), sub-horizontal, moderately curving–undulating,
moderate–high-amplitude seismic reflections that alternate with packages of highly-disrupted
and/or curved low-amplitude seismic reflections (see yellow lines within pink and purple units in
Figure 3a and Figure 4a). The curving geometries of the moderate–high amplitude reflections
display a typical kilometer- to hundreds of meter-scale wavelength and are commonly asymmetric,
seemingly leaning/verging towards the south/SSW (see yellow lines in Figure 4b). Based on ties
with well bores on Edgeøya (Raddedalen-1 well; Bro and Shvarts, 1983; Harland and Kelly, 1997),
these asymmetric, undulate features most likely correspond to SSW-verging folds in Precambrian–
lower Paleozoic basement rocks. In places, apparent reverse offsets of these undulate reflections
align along moderately–gently north- to NNE-dipping surfaces (see red lines in Figure 4a and c),
which are therefore interpreted as minor, top-south/SSW, brittle thrusts.
Upper Paleozoic–Mesozoic sedimentary successions

In Storfjorden and the northwestern Norwegian Barents Sea, shallow (0–3 seconds TWT)

seismic reflections above folded and thrust Precambrian–lower Paleozoic basement rocks show
significantly more continuous patterns (>> five kilometers), gently curving–undulating geometries
and only local disruptions by shallow, dominantly NNE-dipping, high-angle listric disruptions (see
yellow lines within orange unit in Figure 3a and c). In the northeastern Barents Sea, these
reflections are largely flat-lying (see yellow lines within orange unit in Figure 3b and d). Based on
field mapping campaigns and well-bores in adjacent onshore areas of Spitsbergen, Edgeøya, Hopen
and Bjørnøya (see location in Figure 1b), these continuous reflections are interpreted as mildly
folded upper Paleozoic–Mesozoic (–Cenozoic?) sedimentary strata (Dallmann and Krasil'scikov,
1996; Harland and Kelly, 1997; Worsley et al., 2001; Dallmann, 2015). The Permian–Triassic





boundary was correlated throughout the northern Norwegian Barents Sea and Storfjorden by using
the tie of Anell et al. (2014) to the Hopen-2 well.
Deep thrust systems

The packages of sub-horizontal, moderately curving–undulating (folded Precambrian–

lower Paleozoic basement) reflections alternate laterally from north to south with 20–60 kilometers
wide, up to four seconds thick (TWT), upwards-thickening, wedge-shaped packages (areas with
high concentrations of black lines in Figure 3a and d). These wide upwards-thickening packages
consist of two types of reflections. First, they include planar, continuous, gently–moderately north-
to NNE-dipping, sub-parallel, high-amplitude reflections that commonly merge together
downwards and that can be traced and correlated on several seismic sections in Storfjorden (black
lines in Figure 3a). Upwards, these reflections terminate against high-amplitude convex-upwards
reflections interpreted as intra- Precambrian–lower Paleozoic basement reflections (fuchsia lines
in Figure 3a and c) or continue as moderately NNE-dipping disruption surfaces that offset these
intra-basement reflections top-SSW (e.g., offset intra-Precambrian unconformities in Figure 3a and
c and Figure 4d).

Second, sub-parallel, high-amplitude reflections bound wedge-shaped, upwards-thickening

packages of asymmetric, curved, south- to SSW-leaning, moderately north- to NNE-dipping,
moderate-amplitude reflections showing narrow (< one kilometer wide) upwards-convex
geometries (Figure 4d). These asymmetric reflections also commonly appear as gently north- to
NNE-dipping packages of Z-shaped reflections bounded by sub-parallel, planar, high-amplitude
reflections (see yellow lines in Figure 4e). Asymmetric, south- to SSW-leaning, convex-upwards
reflections are interpreted as south- to SSW-verging fold anticlines reflecting relatively low
amounts of plastic deformation of layered rocks.

The alternation of packages of layered rocks folded into SSW-verging folds (yellow lines

in Figure 3a–c) with packages of planar, NNE-dipping, sub-parallel, high-amplitude reflections
(black lines in Figure 3a–c) suggest that the latter reflection packages represent zones where initial
layering was destroyed and/or possibly reoriented, i.e., areas that accommodated larger amounts of
deformation and tectonic displacement. Thus, planar, gently–moderately north- to NNE-dipping,
high-amplitude reflections (black lines in Figure 3a–c) are interpreted as low-angle brittle–ductile
thrust systems. We name these thrust systems (from north to south) the Steiløya–Krylen,





Kongsfjorden–Cowanodden, Bellsundbanken, and Kinnhøgda–Daudbjørnpynten fault zones
(Figure 3a and supplement S2a–b; see Figure 1c for location of the thrusts).

The relatively high-amplitude character of planar, NNE-dipping reflections within the

thrusts suggest that these tectonic structures consist of sub-parallel layers of rocks and minerals
with significantly different physical properties. A probable explanation for such laterally
continuous and consistently high-amplitude reflections is partial recrystallization of rocks layers–
mineral bands into rocks and minerals with significantly higher density along intra-thrust planes
that accommodated large amounts of displacement (e.g., mylonitization; Fountain et al., 1984;
Hurich et al., 1985). In places, packages of aggregates of Z-shaped reflections bounded upwards
and downwards by individual low-angle thrust surfaces are interpreted as forward-dipping duplex
structures (e.g., Boyer and Elliott, 1982) reflecting relatively strong plastic deformation between
low-angle, brittle–ductile (mylonitic?) thrusts (see yellow lines in Figure 4e).

The Kongsfjorden–Cowanodden, Bellsundbanken, and Kinnhøgda–Daudbjørnpynten fault

zones can be traced east-southeast of Edgeøya as a similar series of 20–60 kilometers wide, up to
four seconds thick (TWT), upwards-thickening packages (e.g., black lines in Figure 3d and
supplements S2a). However, their imaging along NNW–SSE-trending seismic sections is much
more chaotic and it is more difficult to identify smaller structures (like south-verging folds and
minor thrusts) within each thrust system (e.g., supplement S2a). This suggests that these three thrust
systems strike oblique to NNW–SSE-trending seismic sections (supplement S2a), whereas they are
most likely sub-orthogonal to N–S- to NNE–SSW-trending seismic sections in Storfjorden (Figure
3a). The only orientation that reconciles these seismic facies variations (i.e., well-imaged on NNE–
SSW-trending seismic sections and poorly imaged by NNW–SSE-trending seismic sections;
Figure 3a and supplement S2a) is an overall WNW–ESE strike.

South of each 20–60 kilometers wide packages of thrust surfaces and related fold and

duplex structures, seismic reflections representing Precambrian–lower Paleozoic basement rocks
typically appear as gently curved, convex-upwards, relatively continuous reflections showing sub-
horizontal seismic onlaps (see white arrows in Figure 3a–f). This suggests that Precambrian–lower
Paleozoic basement rocks most likely consist (meta-) sedimentary rocks (analogous to those
observed in northeastern Spitsbergen and Nordaustlandet; Harland et al., 1993; Stouge et al., 2011)
that were deposited in foreland and piggy-back basins ahead of each 20–60 kilometers wide
packages (Figure 3a–f).





Hence, based on the upwards-thickening geometry of the packages of south- to SSW-
verging folds and of forward-dipping duplexes, on the top-SSW reverse offsets of intra-basement
reflections by low-angle brittle–ductile thrust surfaces, on the upwards truncation of these low-
angle thrusts by intra-basement reflections, and on the onlapping geometries of (meta-)
sedimentary basement rocks south of each set of top-SSW thrust surfaces, the 20–60 kilometers
wide, upwards-thickening, wedge-shaped packages are interpreted as crustal-scale, several
kilometers thick, north- to NNE-dipping, top-SSW, brittle–ductile thrust systems (see fault zones
with high concentration of black lines in Figure 3a–f). These thrust systems include low-angle,
brittle–ductile, mylonitic thrust surfaces (black lines in Figure 4d–e) separating upwards-
thickening thrust sheets that consist of gently to strongly folded basement rocks and forward-
dipping duplex structures (yellow lines in Figure 4d–e). These thrust sheets are interpreted to reflect
accretion and stacking from the north or north-northeast. The interpreted thrust systems are
comparable in seismic facies and thickness to kilometer-thick mylonitic shear zones in the
Norwegian North Sea (Phillips et al. 2016) and southwestern Norwegian Barents Sea (Koehl et al.,

2018).

N–S-trending folds
On E–W seismic cross sections, reflections of the Kongsfjorden–Cowanodden,
Bellsundbanken, and Kinnhøgda–Daudbjørnpynten fault zones define large, 50–100 kilometers
wide, U-shaped, symmetrical depressions (black lines in Figure 3b) on the edge of which they are
truncated at a high angle and overlain by folded lower Paleozoic and mildly folded to flat-lying
upper Paleozoic (meta-) sedimentary rocks (purple and orange units with associated yellow lines
in Figure 3b). In addition, within these U-shaped depressions, the thrust systems show curving up
and down, symmetrical geometries with 5–15 kilometers wavelength (yellow lines within the pink
unit in Figure 3b and Figure 4f). Also notice the kilometer- to hundreds of meter-scale undulating
pattern of 5–15 kilometers wide curved geometries (yellow lines in Figure 4f). Based on the
truncation and abrupt upward disappearance of high-amplitude seismic reflections characterizing
the thrust systems, the high-angle truncation of the thrusts is interpreted as a major erosional
unconformity (dark blue line in Figure 3b and pink line in Figure 4f), and the large U-shaped
depressions as large N–S- to NNE–SSW-trending, upright regional folds (black lines in Figure 3b).
Furthermore, the 5–15 kilometers wide, symmetrical, curved geometries and associated, kilometer-
to hundreds of meter-scale, undulating pattern of seismic reflections within the thrusts are





interpreted as similarly (N–S- to NNE–SSW-) trending, upright, parasitic macro- to meso-scale
folds (yellow lines in Figure 3b and Figure 4f).
Shallow brittle faults

In places, near the top of the 20–60 kilometers wide thrust systems (Kongsfjorden–

Cowanodden, Bellsundbanken, and Kinnhøgda–Daudbjørnpynten fault zones), low-angle brittle–
ductile thrust surfaces merge upwards with high-angle to vertical, listric, north- to NNE-dipping
disruption surfaces at depths of c. 2–3 seconds (TWT; see red lines in Figure 3a and d). These
listric disruption surfaces truncate shallow, laterally continuous reflections that display gently
curved, symmetric geometries in Storfjorden (yellow lines in Figure 3a) and flat-lying geometries
in the northeastern Norwegian Barents Sea (yellow lines in Figure 3d). Notably, they show minor,
down-NNE normal offsets, and related minor southwards thickening (towards the disruption) of
seismic sub-units within Devonian–Carboniferous (–Permian?) sedimentary strata in the north,
both in Storfjorden and the northeastern Barents Sea (Figure 3a–d and white double arrows in
Figure 4g). In addition, they display minor reverse offsets and associated gentle upright folding of
shallow continuous reflections potentially representing upper Mesozoic (–Cenozoic?) sedimentary
deposits in Storfjorden (Figure 3a–c and e, and orange lines in Figure 4h). Note that flat-lying
Mesozoic (–Cenozoic?) sedimentary rocks are not offset in the northeastern Norwegian Barents
Sea (Figure 3d).

Based on the observed normal offsets and southwards-thickening of Devonian–

Carboniferous (–Permian?) sedimentary strata north of these disruption surfaces (e.g., white double
arrows in Figure 4g), these are interpreted as syn-sedimentary Devonian–Carboniferous normal
faults. The minor reverse offsets and associated gentle upright folding of Mesozoic (–Cenozoic?)
sedimentary rocks in Storfjorden (e.g., orange lines in Figure 4h) suggest that these normal faults
were mildly inverted near Svalbard in the Cenozoic. However, it is unclear whether inversion in
Storfjorden initiated in the early Cenozoic or later. Nonetheless, minor reverse offset and folding
of the seafloor clearly indicate ongoing inversion along these faults (Figure 3a and c, and Figure
4h). Furthermore, considering the merging relationship between these high-angle listric disruption
surfaces and underlying shear zones (i.e., merging black and red lines in Figure 3a and c–d), we
propose that the formation of Devonian–Carboniferous normal faults was controlled by the crustal-
scale, north- to NNE-dipping (inherited) thrust systems (Kongsfjorden–Cowanodden,
Bellsundbanken, and Kinnhøgda–Daudbjørnpynten fault zones).






*Nordmannsfonna–Sassenfjorden (eastern–central Spitsbergen)*

Deep thrust system and N–S-trending folds

Seismic data from Nordmannsfonna to Sassenfjorden in eastern Spitsbergen (see Figure 1c for location) show reflection packages including both planar, continuous, moderately-dipping high-amplitude reflections and upwards-curving, moderate-amplitude reflections (black and yellow lines in Figure 3e–f). These two sets are similar to reflection packages interpreted as low-angle, brittle–ductile mylonitic thrusts bounding packages of south- to SSW-verging folds in Storfjorden and the northeastern Norwegian Barents Sea (black and yellow lines in Figure 3a and d, and supplement S2a). In addition, they are located at similar depths (> 2 seconds TWT) and seem to align with the Kongsfjorden–Cowanodden fault zone in Storfjorden along a WNW–ESE-trending axis. Hence, we interpret the deep, continuous, high-amplitude reflections in eastern Spitsbergen as the western continuation of the top-SSW Kongsfjorden–Cowanodden fault zone. This thrust can be traced on seismic data as gently NNE-dipping, high-amplitude reflections in Sassendalen and Sassenfjorden–Tempelfjorden (supplement S2c–d), and possibly in Billefjorden (Koehl et al., 2021 in review, their figure 9a–b).

In Nordmannsfonna, the Kongsfjorden–Cowanodden fault zone (black lines in Figure 3e–f) is truncated upwards by the base-Pennsylvanian unconformity (white line in Figure 3e–f; tied to onshore geological maps; Dallmann, 2015) and shows pronounced variations in dip direction, ranging from east-dipping in the east to NNE-dipping in the north and WNW-dipping in the west, which result into a c. 15–20 kilometers wide, north- to NNE-plunging dome-shaped/convex-upwards geometry (black lines in Figure 3e–f). This portion of the thrust system is interpreted to be folded into a major NNE- to north-plunging upright fold, whose 3D geometry was accurately constrained due to good seismic coverage in this area (Figure 1c).

Small-scale structures within the Kongsfjorden–Cowanodden fault zone also show asymmetric folds and internal seismic units terminating upwards with convex-upwards reflections (yellow lines in Figure 3e–f) suggesting top-SSW nappe thrusting in the northern portion of the thrust system. However, on E–W cross sections, seismic data reveal a set of west-verging folds in the east and a more chaotic pattern of symmetrical, dominantly upright folds in the west (yellow lines in Figure 3e) and below a major, high-angle, east-dipping disruption surface (thick red line in Figure 3e) that crosscuts the Kongsfjorden–Cowanodden fault zone.





Shallow brittle faults

The high-angle, east-dipping disruption surface (thick red line in Figure 3e) is associated with minor subvertical to steeply east-dipping disruption surfaces (thin red lines in Figure 3e). This feature shows a major reverse, top-west offset (> 0.5 second TWT) of seismic units and reflections at depth > 0.75 second (TWT; e.g., black lines in Figure 3e), and minor reverse offset (< 0.1 second TWT) and upwards-convex curving of adjacent reflections at depth < 0.75 second (TWT; white line and yellow lines within blue and units in Figure 3e). Since the major disruption coincides with the location of the Agardhbukta Fault (Piepjohn et al., 2019; see Figure 1 for location) and shows a steep inclination near the surface similar to that of the Agardhbukta Fault, it is interpreted as the subsurface expression of this fault. The Agardhbukta Fault offsets the Kongsfjorden–Cowanodden fault zone in a reverse fashion (>0.5 second TWT; black lines in Figure 3e), and terminates upwards within and slightly offsets upper Paleozoic–Mesozoic sedimentary rocks (blue and black units and associated yellow lines in Figure 3e), which were correlated to onshore outcrops in eastern Spitsbergen (Andresen et al., 1992; Haremo and Andresen, 1992; Dallmann, 2015). As a result, these rocks are folded into a N–S-trending, open, upright fold around the fault tip, both of which suggest top-west movements along the fault (Figure 3e).

Pre-Pennsylvanian dykes

In the hanging wall and on the eastern flank of the folded Kongsfjorden–Cowanodden fault zone in Nordmannsfonna, high- to low-amplitude, gently east-dipping seismic reflections, which possibly represent sedimentary strata (light orange unit in Figure 3e), are crosscut but not offset by moderately west-dipping, high-amplitude planar reflections (blue lines in Figure 3e). In NNE–SSW-trending cross-sections, these high-amplitude, cross-cutting seismic reflections appear sub-horizontal (blue lines in Figure 3f). These crosscutting, west-dipping reflections are mildly folded in places and either terminate upwards within the suggested, gently east-dipping, sedimentary strata (light orange unit in Figure 3e) or are truncated by the base-Pennsylvanian unconformity (white line in Figure 3e). Downwards within the Kongsfjorden–Cowanodden fault zone (black lines in Figure 3e), these inclined reflections can be vaguely traced as a series of discontinuous, subtle features (see blue lines in Figure 3e). In the footwall of the Kongsfjorden–Cowanodden fault zone, the inclined reflections become more prominent again, still do not offset background reflections, and extend to depths of 3–3.5 seconds (TWT; blue lines in Figure 3e). The high amplitude of these planar west-dipping reflections, the absence of offset across them, and their discontinuous





geometries across the Agardhbukta Fault and the Kongsfjorden–Cowanodden fault zone suggest
that they may represent dykes (see Phillips et al., 2018). Because they appear truncated by the Base-
Pennsylvanian unconformity, we suggest such dykes were emplaced prior to development of this
unconformity. The Kongsfjorden–Cowanodden fault zone is folded into a broad, 15–20 kilometers
wide anticline, and offset > 0.5 second (TWT) by the Agardhbukta Fault, whereas the west-dipping
dykes (blue lines in Figure 3e) and the gently east-dipping sedimentary strata they intrude (light
orange unit in Figure 3e) are only mildly folded and show no offset across the Agardhbukta Fault
(Figure 3e). These differences in deformation suggest that the latter were deformed during a mild
episode of late contraction but not by the same early episode of intense contraction that resulted in
macrofolding of the Kongsfjorden–Cowanodden fault zone.
Cretaceous dykes and sills

Near or at the surface, thin, kilometer-wide, lenticular packages of gently dipping,

moderate–high-amplitude seismic reflections (black units in Figure 3e–f) correlate with surface
outcrops of Cretaceous sills of the Diabasodden Suite in eastern Spitsbergen (Senger et al., 2013;
Dallmann, 2015). In places, these sills are associated with areas showing high-frequency
disruptions of underlying sub-horizontal seismic reflections (dotted black lines in Figure 3f)
correlated with onshore occurrences of Pennsylvanian–Mesozoic sedimentary strata (Andresen et
al., 1992; Haremo and Andresen, 1992; Dallmann, 2015). We interpret these areas of high-
frequency disruption in otherwise relatively undisturbed and only mildly deformed Pennsylvanian–
Mesozoic sedimentary strata as zones with occurrences of Cretaceous feeder dykes. Alternatively,
disruption may be related to scattering and attenuation of seismic energy caused on the sills.

*Stappen High (northwestern Norwegian Barents Sea north of Bjørnøya)*

On the Stappen High between Bjørnøya and Spitsbergen (Figure 1c), seismic reflections at

depth of 2–6 seconds (TWT) are dominated by moderate- to high-amplitude reflections with
limited (< five kilometers) lateral continuity showing asymmetric, dominantly SSW-leaning
curving geometries with a few hundreds of meters to a few kilometers width (yellow lines within
pink unit in Figure 3c), i.e., analogous to those in folded Precambrian basement rocks farther north
(Figure 3a and Figure 4a). These reflections are truncated by gently to moderately NNE- (and
subsidiary SSW-) dipping disruption surfaces (black lines within pink and purple units in Figure
3c), some of which connect upwards with shallow (0–2 seconds TWT), NNE-dipping, high-angle





listric disruptions near Bjørnøya in the south (red lines in Figure 3c). Notably, major seismic reflections near the upwards termination of deep, moderately–gently NNE-dipping disruption surfaces display characteristic gently curving-upwards geometries (yellow lines within pink and purple units in Figure 3c) and overlying seismic onlaps (white half arrows in Figure 3c) similar to those observed just south of major NNE-dipping thrust systems in Storfjorden and the northeastern Norwegian Barents Sea (Figure 3a and supplement S2).

We interpret deep (2–6 seconds TWT), curving, discontinuous seismic reflections ((yellow lines within pink and purple units in Figure 3c) as folded Precambrian–lower Paleozoic basement rocks, and dominantly NNE-dipping disruption surfaces (black lines within pink and purple units in Figure 3) as brittle–ductile thrust possibly partly mylonitic, though with less intense deformation than the major NNE-dipping thrust systems observed farther north in Storfjorden and the northeastern Norwegian Barents Sea, like the Kongsfjorden–Cowanodden fault zone. These brittle–ductile thrusts can be traced eastwards on seismic data on the Stappen High and into the Sørkapp Basin (Figure 1c).

Based on their geometries and on gentle folding of the seafloor reflection (yellow lines within green unit in Figure 3c), shallow, NNE-dipping, high-angle listric disruptions are interpreted as mildly inverted normal faults overprinting deep NNE-dipping thrusts. Based on previous fieldwork on Bjørnøya (Worsley et al., 2001), on seismic mapping in the area (Lasabuda et al., 2018), and on well tie to Hopen and Edgeøya, relatively continuous (> five kilometers) shallow (0–2 seconds TWT), gently curved–undulating seismic reflections overlying folded Precambrian–lower Paleozoic basement rocks are interpreted as mildly folded upper Paleozoic–Mesozoic (–Cenozoic?) sedimentary strata (orange and green units in Figure 3c).

***Potential field data and regional gravimetric and magnetic anomalies***

*NNE-dipping thrusts*

In the northern Barents Sea, Storfjorden and central–eastern Spitsbergen, the seismic occurrences of the Kongsfjorden–Cowanodden, Bellsundbanken and Kinnhøgda–Daudbjørnpynten fault zones coincide with gradual, step-like, southwards increases in gravimetry and, in places, with high magnetic anomalies in cross-section (Figure 3a–b and d–f). Similar southwards gradual and step-like increases in the Bouguer and magnetic anomalies correlate with major thrusts north of Bjørnøya (Figure 3c; see Figure 1b for location of Bjørnøya). These patterns



suggest that the footwall of the thrust systems consists of relatively denser rock units, which is
supported by seismic interpretation showing thickening of metamorphosed and folded Precambrian
basement rock units (pink unit in Figure 3a and c–d) in the footwall of the thrusts.

In map-view gravimetric and magnetic data, the three thrust systems in Storfjorden (black

lines in Figure 3a) coincide with three high, WNW–ESE-trending, continuous, gently undulating
(and, in place, merging/splaying) gravimetric and discontinuous magnetic anomalies (dashed
yellow lines in Figure 5a–c) that are separated from each other by areas showing relatively low
gravimetric and magnetic anomalies (e.g., see green to blue areas in Figure 5a). Some of these
anomalies extend from central Spitsbergen to Storfjorden and the northern Barents Sea (below the
Ora and Olga basins) as curving, E–W- and NW–SE-trending, 50–100 kilometers wide anomalies
(dashed yellow lines in Figure 5a–c). Analogously, thrust systems north of Bjørnøya (Figure 3c)
and north of the Ora and Olga basins (supplement S2b) correlate with comparable WNW–ESE-
trending, curving magnetic and gravimetric anomalies (dashed yellow lines in Figure 5a–c). The
WNW–ESE-trending anomalies appear clearer by using a slope-direction shader, which
accentuates the contrast between each trend of anomalies (green and red areas in Figure 5b).

Most of the recognized, regional WNW–ESE-trending magnetic and gravimetric anomalies

(dashed yellow lines in Figure 5a–c) can be traced into the Russian Barents Sea where they are
linear and are crosscut by major N–S- to NNW–SSE-trending anomalies (dashed black and white
lines in Figure 5a–c). Subtle WNW–ESE-trending magnetic and gravimetric anomalies further
extend onshore northwestern Russia (e.g., Kanin Peninsula and southern Novaya Zemlya) where
they correlate with major Timanian thrusts and folds, some of which are suspected to extend
thousands of kilometers between northwestern Russia and the Varanger Peninsula in northern
Norway (e.g., Trollfjorden–Komagelva Fault Zone and Central Timan Fault; Siedlecka, 1975;
Siedlecka and Roberts, 1995; Olovyanishnikov et al., 2000; Kostyuchenko et al., 2006). In addition,
two of the southernmost WNW–ESE-trending gravimetric and magnetic anomalies coincide with
the location of well known, crustal-scale, SSW-verging Timanian thrust faults, the Trollfjorden–
Komagelva Fault Zone and the Central Timan Fault. Thus, based on their overall WNW–ESE
trend, patterns of alternating highs and lows both for gravimetric and magnetic anomalies (see
Figure 5a), location at the boundary of oppositely dipping slopes (see slope-direction shader map
in Figure 5b), and extensive field studies and seismic and well data in northwestern Russia (e.g.,
Kanin Peninsula and Timan Range; Siedlecka and Roberts, 1995; Olovyanishnikov et al., 2000;





Kostyuchenko et al., 2006) and northern Norway (e.g., Varanger Peninsula; Siedlecka, 1975),
WNW–ESE-trending anomalies are interpreted as a combination of basement-seated Timanian
macrofolds and top-SSW reverse faults (Figure 5a–c).

*N–S-trending folds*

Large N–S-trending open folds (e.g., black and yellow lines in Figure 3b) coincide with N–

S- to NNE–SSW-trending, 20–100 kilometers wide, arcuate gravimetric and magnetic anomalies
(dashed white and black lines in Figure 5a–c), which are highly oblique to WNW–ESE-trending
gravimetric and magnetic anomalies and thrust systems (dashed yellow lines in Figure 5a–c).
Notably, major N–S- to NNE–SSW-trending synclines in Figure 3b (marked as red lines over a
white line in Figure 5a and c and as pink lines over a red line in Figure 5b) coincides with similarly
trending gravimetric and magnetic anomalies (dashed black lines in Figure 5a and c and dashed
white lines in Figure 5b). On the slope-direction shader map, these N–S- to NNE–SSW-trending
anomalies are localized along the boundary between areas with eastwards- (ca. 90–100°; blue areas
in Figure 5b) and westwards-facing slopes (ca. 270–280°; white areas in Figure 5b).

Notably where the main thrusts are preserved, major N–S-trending synforms (see 50–60

kilometers wide U-shaped depression formed by the Kinnhøgda–Daudbjørnpynten fault zone, i.e.,
black lines, in Figure 3b) coincide with gravimetric and magnetic highs (white and black dashed
lines in Figure 5a–c), whereas major antiforms where major NNE-dipping thrusts are partly eroded
(e.g., c. 100 kilometers wide areas where the Kinnhøgda–Daudbjørnpynten fault zone is absent in
Figure 3b) coincide with gravimetric and magnetic lows (the lows are parallel to white and black
dashed lines symbolizing magnetic and gravimetric highs in Figure 5a–c). The correlation of the
interpreted NNE-dipping thrust systems with gravimetric highs suggests that the thrusts consist of
relatively denser rocks. This supports the inferred mylonitic component of the thrusts because
mylonites are relatively denser due to the formation of high-density minerals with increasing
deformation (e.g., Arbaret and Burg, 2003; Colombu et al., 2015).

In the northwestern part of the Barents Sea (i.e., area covered by seismic data presented in

Figure 3), N–S- to NNE–SSW-trending gravimetric and magnetic anomalies (white and black
dashed lines in Figure 5a–c) are typically 20–50 kilometers wide and correlate with similarly
trending Caledonian folds and thrusts onshore Nordaustlandet (e.g., Rijpdalen Anticline; Johansson
et al., 2004; 2005; Dumais and Brönner, 2020) and northeastern Spitsbergen (e.g., Atomfjella





Antiform; Gee et al., 1994; Witt-Nilsson et al., 1998), whose width is comparable to that of the
anomalies. In the south, N–S- to NNE–SSW-trending gravimetric and magnetic anomalies merge
together and swing into a NE–SW trend onshore–nearshore the Kola Peninsula and northern
Norway. These anomalies mimic the attitude of Caledonian thrusts and folds in the southern
Norwegian Barents Sea (Gernigon and Brönner, 2012; Gernigon et al., 2014) and onshore northern
Norway (Sturt et al., 1978; Townsend, 1987; Roberts and Williams, 2013). In the east, N–S- to
NNE–SSW-trending anomalies broaden to up to 150 kilometers in the Russian Barents Sea (Figure
5a–c).

In places, the intersections of high, WNW–ESE- and N–S- to NNE–SSW-trending

gravimetric and magnetic anomalies generate relatively higher, oval-shaped anomalies (e.g., dotted
white lines in Figure 5a and c). Notable examples are found in the Ora and Olga basins and east
and south of these basins (see dotted white lines in Figure 5a and c).

**Discussion**

In the discussion, we consider the lateral extent of the interpreted NNE-dipping thrust

systems, their possible timing of formation, and potential episodes of reactivation and overprinting.
Then we briefly discuss the implications of these thrust systems for plate tectonics reconstructions
in the Arctic.

***Extent of NNE-dipping thrust systems***

Four major NNE-dipping systems of mylonitic thrusts and shear zones (Steiløya–Krylen,

Kongsfjorden–Cowanodden, Bellsundbanken, Kinnhøgda–Daudbjørnpynten fault zones) were
identified at depths > 1–2 seconds (TWT) in central–eastern Spitsbergen, Storfjorden and the
northeastern Barents Sea, and several systems with less developed ductile fabrics between
Spitsbergen and Bjørnøya on the Stappen High (Figure 3a–f).

The Kongsfjorden–Cowanodden fault zone is relatively easy to trace and correlate in

Sassenfjorden, Sassendalen, Nordmannsfonna, Storfjorden and the northeastern Barents Sea (east
of Edgeøya) because (i) the seismic data in the these areas have a high resolution and good
coverage, (ii) internal seismic reflections are characterized by high amplitudes (e.g., brittle–ductile
thrusts and mylonitic shear zones), (iii) kinematic indicators within the thrust system consistently
show dominantly top-SSW sense of shear with SSW-verging fold structures (Figure 3a and d–f,





and supplement S2), (iv) the geometry and kinematics indicators along shallow brittle overprints
are regionally consistent (listric, down-NNE, brittle normal faults; Figure 3a and d–f), and (v) this
thrust consistently coincides with increase in gravimetric and magnetic anomaly in cross-section
(Figure 3a and d) and with analogously trending gravimetric and magnetic anomalies in central–
eastern Spitsbergen and the northern Barents Sea (Figure 5a–b). This thrust system was previously
identified below the Ora Basin by Klitzke et al. (2019), though interpreted as potential Timanian
grain instead of a discrete structure. The proposed correlation based on seismic, and cross-section
and map-view gravimetric and magnetic data suggests a lateral extent of c. 550–600 kilometers
along strike for the Kongsfjorden–Cowanodden fault zone. However, the regional magnetic and
gravimetric anomalies associated with this thrust in the Norwegian Barents Sea and Svalbard
extend potentially farther east as a series of WNW–ESE-trending anomalies to the mainland of
Russia (Figure 5a–c). Notably, these anomalies correlate with the southern edge of Novaya Zemlya
(Figure 5a–c) and, more specifically, with WNW–ESE-striking fault segments of the Baidaratsky
fault zone (Figure 1a; Lopatin et al., 2001; Korago et al., 2004), a major thrust fault that bounds a
major basement high in the central Russian Barents Sea, the Ludlov Saddle (Johansen et al., 1992;
Drachev et al., 2010). Thus, it is possible that the Kongsfjorden–Cowanodden fault zone also
extends farther east, possibly merging with the Baidaratsky fault zone, i.e., with a minimum extent
of 1700–1800 kilometers (Figure 5a–c).
The overall NNE-dipping and folded (into NNE-plunging folds) geometry of the
Kongsfjorden–Cowanodden fault zone (Figure 3e–f and Klitzke et al., 2019, their figures 3–5) may
explain the alternating NW–SE- and E–W-trending geometry of the gravimetric and magnetic
anomalies correlating with this thrust system (Figure 5a–b). E–W- and NW–SE-trending segments
of these anomalies may represent respectively the western and eastern limbs of open, gently NNE-
plunging macro-anticlines in the northern Norwegian Barents Sea. This is supported by the
relatively higher, oval-shaped gravimetric and magnetic anomalies at the intersection of WNW–
ESE- and N–S- to NNE–SSW-trending magnetic and gravimetric highs, which are interpreted as
the interaction of two sub-orthogonal fold trends (Figure 5a and c).
Interpretation of seismic sections (Figure 3e–f and supplement S2) and regional magnetic
and gravimetric data (Figure 5a–c) in central–eastern Spitsbergen show that NNE-dipping, top-
SSW Kongsfjorden–Cowanodden and Bellsundbanken fault zones likely extend westwards into
central (and possibly northwestern) Spitsbergen (e.g., Sassendalen, Sassenfjorden, Tempelfjorden,



and Billefjorden; see Figure 1c for locations). This is further supported by recent field, bathymetric
and seismic mapping in central Spitsbergen showing that (inverted) Devonian–Carboniferous
NNE-dipping brittle normal faults in Billefjorden and Sassenfjorden–Tempelfjorden merge with
kilometer-scale, NNE-dipping, Precambrian basement fabrics and shear zones at depth (Koehl,
2020a; Koehl et al., 2021 in review). Other examples of WNW–ESE-trending fabrics include faults
within Precambrian basement and Carboniferous sedimentary rocks in northeastern Spitsbergen
(Witt-Nilsson et al., 1998; Koehl and Muñoz-Barrera, 2018), and within Devonian sedimentary
rocks in northern and northwestern Spitsbergen (Friend et al., 1997; McCann, 2000; Dallmann and
Piepjohn, 2020). These suggest a repeated and regional influence of WNW–ESE-trending thrust
systems and associated basement fabrics in Spitsbergen.
Analogously to the Kongsfjorden–Cowanodden fault zone, the Bellsundbanken and
Kinnhøgda–Daudbjørnpynten fault zones (Figure 3a) geometries and kinematics on seismic data,
and their coinciding with parallel gravimetric and magnetic anomalies in map view and with
magnetic and gravimetric highs in cross-section suggest that they extend from Storfjorden to the
island of Hopen (Figure 1c, Figure 3a, Figure 5a–c, and supplement S2). Notably, a 50–100
kilometers wide, NNE–SSW-trending gravimetric and associated magnetic anomaly interpreted as
Caledonian grain in Nordaustlandet (Rijpdalen Anticline; Dumais and Brönner, 2020) bends across
the trace of these two thrust systems (Figure 5a–c). Farther east, the Bellsundbanken and
Kinnhøgda–Daudbjørnpynten fault zones parallel gravimetric and magnetic, alternating E–W- and
NW–SE-trending anomalies that follow the trends and map-view shapes of the Ora and Olga basins
in the northeastern Norwegian Barents Sea (Anell et al., 2016; see Figure 1b–c for location). This
suggests that these two thrust systems extend into the northeastern Norwegian Barents Sea and,
potentially, into the Russian Barents Sea, and affected the development of Paleozoic sedimentary
basins. This is also the case of the Steiløya–Krylen fault zone (supplement S2b), which coincides
with mild, discontinuous, WNW–ESE-trending gravimetric and magnetic anomalies that extend
well into the Russian Barents Sea and, possibly, across Novaya Zemlya (Figure 5a–c).
In southwestern Spitsbergen, field mapping revealed the presence of a major, subvertical,
kilometer-thick, WNW–ESE-striking mylonitic shear zone metamorphosed under amphibolite
facies conditions, the Vimsodden–Kosibapasset Shear Zone (Majka et al., 2008, 2012; Mazur et
al., 2009; see Figure 1c for location). This major sinistral shear zone aligns along a WNW–ESE-
trending axis with the Kinnhøgda–Daudbjørnpynten fault zone in the northwestern Norwegian





Barents Sea (Figure 3a), and shows a folded geometry in map view that is comparable to that of
major NNE-dipping thrust systems in the northern Norwegian Barents Sea (Figure 3a and e–f,
Figure 5a–c, and supplement S2; Klitzke et al., 2019). In addition, the Vimsodden–Kosibapasset
Shear Zone juxtaposes relatively old Proterozoic basement rocks in the north against relatively
young rocks in the south, thus suggesting a similar configuration and kinematics as along the
Kinnhøgda–Daudbjørnpynten fault zone in Storfjorden and the northeastern Norwegian Barents
Sea. Moreover, von Gosen and Piepjohn (2001) and Bergh and Grogan (2003) reported that
Devonian–Mississippian sedimentary successions and Cenozoic fold structures (e.g., Hyrnefjellet
Anticline) are offset sinistrally by a few kilometers in Hornsund. Thus, we propose that the
Vimsodden–Kosibapasset Shear Zone extends into Hornsund and represents the westwards
continuation of the Kinnhøgda–Daudbjørnpynten fault zone. This suggests a minimum extent of
400–450 kilometers for this thrust system (Figure 1b–c and Figure 5a–c).

***Timing of formation of major NNE-dipping thrust systems and N–S-trending folds***
*NNE-dipping thrust systems*

The several-kilometer thickness and hundreds–thousands of kilometers along-strike extent

of NNE-dipping thrust systems in central–eastern Spitsbergen, Storfjorden, and the northwestern
and northeastern Norwegian Barents Sea suggest that they formed during a major contractional
tectonic event. The overall WNW–ESE trend and the consistent north-northeastwards dip and top-
SSW sense of shear along the newly evidenced deep thrust systems preclude formation during the
Grenvillian, Caledonian, and Ellesmerian orogenies, and the Eurekan tectonic event. These tectonic
events all involved dominantly E–W-oriented contraction and resulted in the formation of overall
N–S- to NNE–SSW-trending fabrics, structures and deformation belts in Svalbard (i.e., sub-
orthogonal to the newly identified thrust systems) such as the Atomfjella Antiform (Gee et al.,
1994; Witt-Nilsson et al., 1998), the Vestfonna and Rijpdalen anticlines (Johansson et al., 2004;
2005; Dumais and Brönner, 2020), the Dickson Land and Germaniahalvøya fold-thrust zones
(McCann, 2000; Piepjohn, 2000; Dallmann and Piepjohn, 2020), and the West Spitsbergen Fold-
and-Thrust Belt and related early Cenozoic structures in eastern Spitsbergen (Andresen et al., 1992;
Haremo and Andresen, 1992; Dallmann et al., 1993), and NE–SW- to NNE–SSW-striking thrusts
and folds in northern Norway (Sturt et al., 1978; Townsend, 1987; Roberts and Williams, 2013)
and the southwestern Barents Sea (Gernigon et al., 2014).



A possible cause for the formation of the observed NNE-dipping thrust systems is the late Neoproterozoic Timanian Orogeny, which is well known onshore northwestern Russia (e.g., Kanin Peninsula, Timan Range and central Timan; Siedlecka and Roberts, 1995; Olovyanishnikov et al., 2000; Kostyuchenko et al., 2006) and northeastern Norway (Varanger Peninsula; Siedlecka and Siedlecki, 1967; Siedlecka, 1975; Roberts and Olovyanishnikov, 2004), and traces of which were recently found in southwestern Spitsbergen (Majka et al., 2008, 2012, 2014) and northern Greenland (Rosa et al., 2016; Estrada et al., 2018). The overall transport direction during this orogeny was directed towards the south-southwest and most thrust systems show NNE-dipping geometries (Olovyanishnikov et al., 2000; Kostyuchenko et al., 2006), e.g., the Timanian thrust front on the Varanger Peninsula in northeastern Norway (Trollfjorden–Komagelva Fault Zone; Siedlecka and Siedlecki, 1967; Siedlecka, 1975). In addition, the size of Timanian thrust systems in the Timan Range (e.g., Central Timan Fault) is comparable ($\geq$ 3–4 seconds TWT; Kostyuchenko et al., 2006 their figure 17) to that of thrust systems in the northern Norwegian Barents Sea and Svalbard (Figure 3a and c–d).

Thus, based on their overall WNW–ESE strike (Figure 1b–c), their vergence to the south-southwest (Figure 3a, c–d and f), their coincidence with gravimetric and magnetic highs (Figure 5a–c), their upward truncation by a major unconformity consistently throughout the study area (see top-Precambrian unconformity in Figure 3a–d), and the correlation of these NNE-dipping thrusts (via gravimetric and magnetic anomalies) to similarly striking and verging structures of comparable size (i.e., several seconds TWT thick) onshore–nearshore northwestern Russia and northern Norway (Siedlecka, 1975; Siedlecka and Roberts, 1995; Olovyanishnikov et al., 2000; Roberts and Siedlecka, 2002; Gee and Pease, 2004; Kostyuchenko et al., 2006), NNE-dipping thrusts in the northern Norwegian Barents Sea, Storfjorden, and central–eastern Spitsbergen are interpreted as the western continuation of Timanian thrusts.

Timanian grain was recently identified in the northeastern Norwegian Barents Sea through interpretation of new seismic, magnetic and gravimetric datasets shown in Figure 5a–c (Klitzke et al., 2019). The alignment, coincident location, and matching geometries (e.g., curving E–W to NW–SE strike/trend and kilometer-wide NNE–SSW-trending anticline) between Timanian grain and structures mapped by Klitzke et al. (2019) and the major, NNE-dipping, top-SSW thrust systems described in central–eastern Spitsbergen, Storfjorden and the Norwegian Barents Sea (Figure 3a–f and supplement S2) further support a Timanian origin for the latter. Further evidence



of relic Timanian structural grain as far as the Loppa High and Bjørnøya Basin are documented by
previous magnetic studies and modelling (Marello et al., 2010). Moreover, seismic mapping
suggests that Timanian thrust systems extend well into central Spitsbergen (Figure 3e–f and
supplement S2c–d; Koehl, 2020a; Koehl et al., 2021 in review), and regional gravimetric and
magnetic anomaly maps suggest that Timanian thrust systems might extend farther west to (north-
) western Spitsbergen (Figure 5a–c).

Probable reasons as to why these major (hundreds–thousands of kilometers long) thrust

systems were not identified before during fieldwork in Svalbard are their burial to high depth (>
1–2 seconds TWT in the study area, i.e., several kilometers below the surface; Figure 3a–f), and
their strong overprinting by younger tectonic events like the Caledonian Orogeny in areas where
they are exposed (e.g., Vimsodden–Kosibapasset Shear Zone in southwestern Spitsbergen;
Faehnrich et al., 2020). Possible areas of interest for future studies include the western and
northwestern parts of Spitsbergen where Caledonian and Eurekan E–W contraction contributed to
uplift and exhume deep basement rocks, and where Timanian rocks potentially crop out (e.g.,
Peucat et al., 1989).

*N–S-trending folds*

N–S-trending upright folds involve the NNE-dipping thrust systems (Figure 3b and e) and

correlate (via gravimetric and magnetic anomalies) with major Caledonian folds in northeastern
Spitsbergen and Nordaustlandet, like the Atomfjella Antiform (Gee et al., 1994; Witt-Nilsson et
al., 1998) and Rijpdalen Anticline (Johansson et al., 2004; 2005; Dumais and Brönner, 2020), with
Caledonian grain in the southern Norwegian Barents Sea (Gernigon and Brönner, 2012; Gernigon
et al., 2014), and with major NE–SW-trending Caledonian folds onshore northern Norway (Sturt
et al., 1978; Townsend, 1987; Roberts and Williams, 2013). In addition, the width of the NE–SW-
to N–S-trending gravimetric and magnetic anomalies associated with these folds increases up to
150 kilometers eastwards, i.e., away from the Caledonian collision zone (Figure 5a–c; Corfu et al.,
2014; Gasser, 2014). Thus, N–S-trending folds in the northern Norwegian Barents Sea are
interpreted as Caledonian regional folds in Precambrian–lower Paleozoic rocks. The relatively
broader geometry of Caledonian folds away from the Caledonian collision zone (e.g., in the
Russian Barents Sea) is inferred to be related to gentler fold geometries due to decreasing
deformation intensity in this direction. This is further supported by relatively low grade Caledonian



metamorphism in Franz Josef Land (Knudsen et al., 2019; see Figure 1a–b for location). By contrast, the presence of tighter Caledonian folds near the collision zone in the northern Norwegian Barents Sea (e.g., Figure 3b and e, and Atomfjella Antiform and Rijpdalen Anticline onshore; Gee et al., 1994; Witt-Nilsson et al., 1998; Johansson et al., 2004, 2005; Dumais and Brönner, 2020) is associated with much narrower (20–50 kilometers wide) gravimetric and magnetic anomalies (Figure 5a–c). Note that the Atomfjella Antiform and Rijpdalen Anticline can be directly correlated with 20–50 kilometers wide, N–S-trending high gravimetric and magnetic anomalies (Figure 5a–c). Noteworthy, some of the NNE–SSW-trending folds and anomalies in the northernmost Norwegian Barents Sea may reflect a combination of Caledonian and superimposed early Cenozoic Eurekan folding (e.g., Kairanov et al., 2018).

The interference of WNW–ESE- and N–S- to NNE–SSW-trending gravimetric highs, which are correlated to Timanian and Caledonian folds respectively, produces oval-shaped gravimetric and magnetic highs (Figure 5a). These relatively higher, oval-shaped gravimetric anomalies are interpreted to correspond to dome-shaped folds resulting from the interaction of Timanian and Caledonian folds involving refolding of WNW–ESE-trending Timanian folds during E–W Caledonian contraction. This interpretation is supported by field studies on the Varanger Peninsula in northern Norway and by seismic studies of Timanian thrusts off northern Norway where the interaction of Timanian and Caledonian folds produced dome-shaped fold structures (Ramsay, 1962), e.g., like the Ragnarokk Anticline (Siedlecka and Siedlecki, 1971; Koehl, in prep.). Furthermore, Barrère et al. (2011) suggested that basins and faults in the southern Norwegian Barents Sea are controlled by the interaction of Caledonian and Timanian structural grain, and Marello et al. (2010) argued that elbow-shaped magnetic anomalies reflect the interaction of Caledonian and Timanian structural grains in the Barents Sea, potentially as far west as the Loppa High and the Bjørnøya Basin.

***Phanerozoic reactivation and overprinting of Timanian thrust systems***

*Caledonian reactivation and overprint*

The geometry of the Kongsfjorden–Cowanodden and Kinnhøgda–Daudbjørnpynten fault zones in Nordmannsfonna (Figure 3e) and the northeastern Norwegian Barents Sea (Figure 3b; Klitzke e a., 2019), where they are folded into broad NNE-plunging upright anticlines and synclines suggests that these thrust systems were deformed after they accommodated top-SSW





Timanian thrusting (Figure 6a and Figure 7a). In addition, subsidiary top-west kinematics (west-verging folds and top-west minor thrusts) suggest that these thrust systems were partly reactivated–overprinted during an episode of intense E–W contraction (Figure 6b and Figure 7b). However, west-dipping dykes crosscutting and gently east-dipping sedimentary strata overlying the eastern part of the folded Kongsfjorden–Cowanodden fault zone are only mildly folded, and upper Paleozoic sedimentary strata lie flat over folded and partly eroded Precambrian–lower Paleozoic rocks and the Kinnhøgda–Daudbjørnpynten fault zone, thus suggesting that these sedimentary strata and dykes were not involved in this episode of E–W contraction (Figure 3e).

A notable episode of E–W contraction in Svalbard is the Caledonian Orogeny in the early–mid Paleozoic, which resulted in the formation of west-verging thrusts and N–S-trending folds of comparable size (c. 15–25 kilometers wide) to those affecting the Kongsfjorden–Cowanodden and Kinnhøgda–Daudbjørnpynten fault zones in Nordmannsfonna and the northern Norwegian Barents Sea (Figure 3b and e; Klitzke et al., 2019, their figures 3–5), such as the Atomfjella Antiform in northeastern Spitsbergen (Gee et al., 1994; Witt-Nilsson et al., 1998; Lyberis and Manby, 1999) and the Rijpdalen Anticline in Nordaustlandet (Figure 1b). Since the NNE-plunging anticline in Nordmannsfonna does not affect overlying Pennsylvanian–Mesozoic sedimentary strata (Figure 3e), we propose that they formed during Caledonian contraction (Figure 7b). This is supported by the involvement of the top-Precambrian unconformity and underlying NNE-dipping thrusts in N–S- to NNE-SSW-trending folds, and by the truncation of these folds by the top-Silurian unconformity, which is onlapped by mildly deformed to flat-lying upper Paleozoic strata (Figure 3b and Figure 4f). Furthermore, structures with geometries comparable to NNE-plunging folds in the northern Barents Sea and Svalbard were observed in northern Norway. An example is the Ragnarokk Anticline, a dome-shaped fold structure along the Timanian front thrust on the Varanger Peninsula, which results from the re-folding of Timanian thrusts and folds into a NE–SW-trending Caledonian trend (Siedlecka and Siedlecki, 1971).

Further support of a Caledonian origin for upright NNE-plunging folds in eastern Spitsbergen, Storfjorden and the northern Norwegian Barents Sea is that these folds are relatively tight in the west, in Nordmannsfonna and the northwestern Barents Sea (Figure 3b and e), whereas they show gradually gentler and more open geometries in the east, i.e., away from the Caledonian collision zone (Figure 3b). This is also shown by the gradual eastwards broadening of regional gravimetric and magnetic anomalies correlated with Caledonian folds suggesting gentler fold



geometries related to decreasing (Caledonian) deformation intensity in this direction (Figure 5a–
c). This contrasts with the homogeneous intensity of deformation along NNE-dipping thrusts on
seismic data and with the homogeneous width of related gravimetric–magnetic anomalies from
west to east in Svalbard and the Barents Sea (Figure 3a–f and Figure 5a–c and supplement S2).

In Nordmannsfonna, the Caledonian origin of the major 15–20 kilometers wide anticline,

and the truncation of overlying, gently east-dipping, mildly folded sedimentary strata and
crosscutting west-dipping dykes by the base-Pennsylvanian unconformity suggest that these
sedimentary strata and dykes are Devonian (–Mississippian?) in age (Figure 6c–d). This is
supported by the presence of thick Devonian–Mississippian collapse deposits in adjacent areas of
central–northern Spitsbergen (Cutbill et al., 1976; Murascov and Mokin, 1979; Aakvik, 1981;
Gjelberg, 1983; Manby and Lyberis, 1992; Friend et al., 1997), and by Middle Devonian to
Mississippian ages (395–327 Ma) for dykes in central–northern Spitsbergen (Evdokimov et al.,
2006), northern Norway (Lippard and Prestvik, 1997; Guise and Roberts, 2002), and northwestern
Russia (Roberts and Onstott, 1995).

The occurrence of a > 0.5 second (TWT) reverse offset of the folded Kongsfjorden–

Cowanodden fault zone and the lack of offset of the Devonian (–Mississippian?) dykes across the
Agardhbukta Fault indicate that the latter fault formed as a top-west thrust during the Caledonian
Orogeny. At depth, the Agardhbukta Fault merges with the eastern flank of the folded
Kongsfjorden–Cowanodden fault zone. This, together with the presence of minor, high-angle, top-
west brittle thrusts within the Kongsfjorden–Cowanodden fault zone (Figure 3e), indicates that the
Agardhbukta Fault reactivated and/or overprinted the eastern portion of the Kongsfjorden–
Cowanodden fault zone in Nordmannsfonna during Caledonian contraction (Figure 6b and Figure
7b). Depth conversion using seismic velocities from Gernigon et al. (2018) suggest that the
Agardhbukta Fault offset the Kongsfjorden–Cowanodden fault zone by ca. 2.4–2.5 kilometers top-
west during Caledonian contraction (Figure 3e and supplement S3g). These kinematics are
consistent with field observation in eastern Spitsbergen by Piepjohn et al. (2019, their figure 17b).
However, Piepjohn et al. (2019) also suggested a significant component of Mesozoic–Cenozoic,
down-east normal movement, which was not identified in Nordmannsfonna. This suggests either
along strike variation in the movement history of the Agardhbukta Fault, either that the fault
mapped on seismic data in Nordmannsfonna does not correspond to the Agardhbukta Fault of
Piepjohn et al. (2019).



Considering the presence of crustal-scale, NNE-dipping, hundreds (to thousands?) of
kilometers long (Timanian) thrust systems extending from the Barents Sea (and possibly from
onshore Russia) to central–eastern and southern Spitsbergen and the northwestern Norwegian
Barents Sea (Figure 5a–c) prior to the onset of E–W-oriented Caledonian contraction, it is probable
that such large structures would have (at least partially) been reactivated and/or overprinted during
subsequent tectonic events if suitably oriented. Under E–W contraction, WNW–ESE-striking,
dominantly NNE-dipping Timanian faults would be oriented at c. 30° to the direction of principal
stress and, therefore, be suitable (according to Anderson's stress model) to reactivate/be
overprinted with sinistral strike-slip movements. Such kinematics were recorded along the
Vimsodden–Kosibapasset Shear Zone in Wedel Jarlsberg Land (Mazur et al., 2009) and within
Hornsund (von Gosen and Piepjohn, 2001).
However, recent $^{40}$Ar–$^{39}$Ar geochronological determinations on muscovite within this
structure suggest that this structure formed during the Caledonian Orogeny (Faehnrich et al. 2020).
Nonetheless, the same authors also obtained Timanian ages (600–540 Ma) for (initial) movements
along minor shear zones nearby and parallel to the Vimsodden–Kosibapasset Shear Zone. Since
this large shear zone must have represented a major preexisting zone of weakness when Caledonian
contraction initiated, it is highly probable that it was preferentially chosen to reactivate instead of
minor shear zones. Thus, the Caledonian ages obtained along the Vimsodden–Kosibapasset Shear
Zone most likely reflect complete resetting of the geochronometer along the shear zone due to large
amounts of Caledonian reactivation–overprinting, while minor nearby shear zones preserved traces
of initial Timanian deformation. This is also supported by observations in northern Norway
suggesting that Timanian thrusts (e.g., Trollfjorden–Komagelva Fault Zone) were reactivated as
major strike-slip faults during the Caledonian Orogeny (Roberts, 1972; Herrevold et al., 2009; Rice
2014). This interpretation reconciles the strong differences in dipping angle and depth between the
Kinnhøgda–Daudbjørnpynten fault zone and the Vimsodden–Kosibapasset Shear Zone. The
former was located away from the Caledonian collision zone and essentially retained its initial,
moderately NNE-dipping Timanian geometry and was deeply buried during the Phanerozoic,
whereas the latter was intensely deformed, pushed into a sub-vertical position, and uplifted an
exhumed to the surface because it was located near or within the Caledonian collision zone.

*Devonian–Carboniferous normal overprint–reactivation*



In Nordmannsfonna, the wedge shape of Devonian (–Mississippian?) sedimentary strata in the hanging wall of the Kongsfjorden–Cowanodden fault zone suggest that the eastern portion of this thrust was reactivated as a gently–moderately dipping extensional detachment (Figure 6c) and, thus, that Devonian (–Mississippian?) strata in this area represent analogs to collapse deposits in northern Spitsbergen. This is supported by the intrusion of west-dipping Devonian (–Mississippian?) dykes orthogonal to the eastern portion of the thrust system, i.e., orthogonal to extensional movements along the inverted east-dipping portion of the thrust (Figure 3e and Figure 6d). Similar relationships were inferred in northwestern Spitsbergen, where Devonian collapse sediments were deposited along a N–S-trending Precambrian basement ridge bounded by a gently dipping, extensional mylonitic detachment (Braathen et al., 2018).

In Sassenfjorden, Storfjorden and the northeastern Norwegian Barents Sea, listric brittle normal faults showing down-NNE offsets and syn-tectonic thickening within Devonian–Carboniferous (–Permian?) sedimentary strata merge at depth with the uppermost part of NNE-dipping Timanian thrust systems like the Kongsfjorden–Cowanodden fault zone (Figure 3a and d and supplement S2c). This indicates that Timanian thrust systems were used as preexisting zones of weakness during late–post-orogenic collapse of the Caledonides in the Devonian–Carboniferous (Figure 6c–e and Figure 7c).

The presence of the Kongsfjorden–Cowanodden fault zone in Storfjorden and below Edgeøya also explains the strong differences between the Paleozoic sedimentary successions penetrated by the Plurdalen-1 and Raddendalen-1 exploration wells (Bro and Shvarts, 1983; Harland and Kelly, 1997). Notably, the Plurdalen-1 well penetrated (at least) ca. 1600 meters thick Devonian–Mississippian sedimentary rocks in the direct hanging wall of the Kongsfjorden–Cowanodden fault zone and related listric brittle overprints (Figure 3a), whereas the interpretation of Bro and Shvarts (1983) suggests that the Raddedalen-1 well encountered thin (90–290 meters thick) Mississippian strata overlying (> 2 kilometers) thick lower Paleozoic sedimentary rocks ca. 30 kilometers farther northeast, i.e., away from the Kongsfjorden–Cowanodden fault zone and related overprints. The presence of thick Devonian sedimentary strata in the direct hanging wall of listric overprints of the Kongsfjorden–Cowanodden fault zone further supports late–post-Caledonian extensional reactivation–overprinting of NNE-dipping Timanian thrusts.

In central Spitsbergen, recently identified Early Devonian–Mississippian normal faults formed along and overprinted–reactivated major NNE-dipping ductile (mylonitic) shear zones and



fabrics in Billefjorden (Koehl et al., 2021 in review) and Sassenfjorden–Tempelfjorden (Koehl,
2020a). These show sizes, geometries and kinematics comparable to those of the Kongsfjorden–
Cowanodden fault zone, and are, therefore, interpreted as the western continuation of this thrust
system. The Devonian–Carboniferous extensional reactivation–overprinting of the Kongsfjorden–
Cowanodden fault zone in central Spitsbergen explains the southward provenance of northwards
prograding sedimentary rocks of the uppermost Silurian–Lower Devonian Siktefjellet and Red Bay
groups and Wood Bay Formation and the enigmatic WNW–ESE trend of the southern boundary of
the Devonian Graben in central–northern Spitsbergen (Gee and Moody-Stuart, 1966; Friend et al.,
1966; Friend and Moody-Stuart, 1972; Murascov and Mokin, 1979; Friend et al., 1997; McCann,
2000; Dallmann and Piepjohn, 2020; Koehl et al., 2021 in review).

*Mild Triassic overprint*

The Kongsfjorden–Cowanodden fault zone and associated overprints align with WNW–

ESE- to NW–SE-striking normal faults onshore southern and southwestern Edgeøya in
Kvalpynten, Negerpynten, and Øhmanfjellet (Osmundsen et al., 2014; Ogata et al., 2018). These
faults display both listric and steep planar geometries in cross-section and bound thickened syn-
sedimentary growth strata in lowermost Upper Triassic sedimentary rocks of the Tschermakfjellet
and De Geerdalen formations (Ogata et al., 2018; Smyrak-Sikora et al., 2020). The Norwegian
Barents Sea and Svalbard are believed to have remained tectonically quiet throughout the Triassic
apart from minor deep-rooted normal faulting in the northwestern Norwegian Barents Sea (Anell
et al., 2013) and Uralides-related contraction in the (south-) east (Müller et al., 2019). Hence, we
propose that the progradation and accumulation of thick sedimentary deposits of the Triassic deltaic
systems above the southeastward continuation of the Kongsfjorden–Cowanodden fault zone may
have triggered minor tectonic adjustments resulting in the development of a system of small half-
grabens over the thrust system. Alternatively or complementary, the deposition of thick Triassic
deltaic systems may have locally accelerated compaction of sedimentary strata underlying the
Tschermakjellet Formation in south- and southwest-Edgeøya, e.g., of the potential pre-Triassic
syn-tectonic growth strata along the Kongsfjorden–Cowanodden fault zone, and, thus, facilitated
the development of minor half-grabens within the Triassic succession along this thrust system.

*Eurekan reactivation–overprint*





In eastern Spitsbergen, the Agardhbukta Fault segment of the Lomfjorden Fault Zone

truncates the Kongsfjorden–Cowanodden fault zone with a major, > 0.5 second (TWT) top-west
reverse offset (Figure 3e). The Agardhbukta fault also mildly folds Pennsylvanian–Mesozoic
sedimentary rocks and Cretaceous sills into a gentle upright (fault-propagation) fold with no major
offset (Figure 6f–g), which is supported by onshore field observations in eastern and northeastern
Spitsbergen (Piepjohn et al., 2019). Mild folding of Mesozoic sedimentary rocks and of Cretaceous
intrusions indicates that the Agardhbukta Fault was most likely mildly reactivated as a top-west
thrust during the early Cenozoic Eurekan tectonic event (Figure 6g and Figure 7d).

Seismic data show that high-angle listric Devonian–Carboniferous normal faults were

mildly reactivated as reverse faults that propagated upwards and gently folded adjacent upper
Paleozoic–Mesozoic (–Cenozoic?) sedimentary strata in the northwestern Norwegian Barents Sea,
Storfjorden and central–eastern Spitsbergen (Figure 3a–c and supplement S2), but not in the
northeastern Norwegian Barents Sea (Figure 3d). Since normal faults were not inverted in the east,
it is probable that inversion of these faults in central–eastern Spitsbergen, Storfjorden and the
northwestern Norwegian Barents Sea first occurred during the Eurekan tectonic event in the early
Cenozoic, when Greenland collided with western Spitsbergen (Figure 7d). This is also supported
by the gently folded character of Devonian–Mesozoic (–Cenozoic?) sedimentary successions in
the west (Figure 3a and c), whereas these successions are essentially flat-lying (i.e., undeformed)
in the east (Figure 3b and d). Nevertheless, folding of the seafloor reflection in Storfjorden and the
northwestern Norwegian Barents Sea suggests ongoing contractional deformation along several of
these faults in the northwestern Norwegian Barents Sea and Storfjorden (Figure 3a–c).

Major, top-SSW mylonitic shear zones in Sassenfjorden–Tempelfjorden and Billefjorden

display early Cenozoic overprints including top-SSW duplexes in uppermost Devonian–
Mississippian coals of the Billefjorden Group acting as a partial décollement along a major
basement-seated listric brittle fault (Koehl, 2020a; supplement S2) and NNE-dipping brittle faults
offsetting the east-dipping Billefjorden Fault Zone by hundreds of meters to several kilometers left-
laterally (Koehl et al., 2021 in review). Thus, the correlation of the Kongsfjorden–Cowanodden
fault zone with these top-SSW mylonitic shear zones in Sassenfjorden–Tempelfjorden and
Billefjorden (see Figure 1c for location) supports reactivation–overprinting of major NNE-dipping
Timanian thrust systems as top-SSW, sinistral-reverse, oblique-slip thrusts in the early Cenozoic
Eurekan tectonic event. Such correlation explains the NW–SE trend and the location of the





northeastern boundary of the Central Tertiary Basin, which terminates just southwest of Sassenfjorden and Sassendalen in central Spitsbergen (Figure 1b–c). It also explains the dominance of NW–SE- to WNW–ESE-striking faults within Cenozoic deposits of the Central Tertiary Basin (Livshits, 1965a), and the northwestwards provenance (Petersen et al., 2016) and northwards thinning of sediments deposited in the basin (Livshits, 1965b), which were probably sourced from uplifted areas in the hanging wall of the reactivated–overprinted thrust.

Noteworthy, Livshits (1965a) argued that the Central Tertiary Basin was bounded to the north by a major WNW–ESE-striking fault extending from Kongsfjorden to southern Billefjorden– Sassenjorden where the NNE-dipping Kongsfjorden–Cowanodden fault zone was mapped (present study; supplement S2). This indicates that the Kongsfjorden–Cowanodden fault zone might extend west of Billefjorden and Sassenfjorden, potentially until Kongsfjorden (see Figure 1c for location). Should it be the case, the Kongsfjorden–Cowanodden fault zone would coincide with a major terrane boundary in Svalbard, which was speculated to correspond to one or more regional WNW– ESE- to N–S-striking faults in earlier works, e.g., Kongsvegen Fault and Lapsdalen Thrust (Harland and Horsfield, 1974), Kongsvegen Fault Zone and/or Central–West Fault Zone (Harland and Wright, 1979), and Kongsfjorden–Hansbreen Fault Zone (Harland et al., 1993). The presence of a major, (inherited Timanian) NNE-dipping, basement-seated fault zone in this area would explain the observed strong differences between Precambrian basement rocks in Svalbard's northwestern and southwestern terranes.

In southern Spitsbergen, von Gosen and Piepjohn (2001) and Bergh and Grogan (2003) suggested the presence of a WNW–ESE-striking, sinistral-reverse strike-slip fault in Hornsund based on a one-kilometer left-lateral offset of Devonian–Carboniferous sedimentary successions and of the early Cenozoic Hyrnefjellet Anticline across the fjord. This fault is part of the Kinnhøgda–Daudbjørnpynten fault zone and was most likely reactivated–overprinted during Eurekan contraction–transpression in the early Cenozoic.

*Present day tectonism*

Seismic data show that the seafloor reflection is folded and/or offset in a reverse fashion by high-angle brittle faults merging at depth with interpreted Timanian thrust systems in Storfjorden and just north of Bjørnøya in the northwestern Norwegian Barents (Figure 3a and c, and Figure 4h). This indicates that some of the Timanian thrust systems are still active at present and are





reactivated/overprinted by reverse faults (Figure 7e). A potential explanation for ongoing
reactivation–overprinting is transfer of extensional tectonic stress in the Fram Strait as ridge-push
tectonism through Spitsbergen and Storfjorden.

***Implication for plate tectonics reconstructions of the Barents Sea and Svalbard in the late***
***Neoproterozoic–Paleozoic***
The presence of hundreds to thousands of kilometers long Timanian faults throughout the
northern Norwegian Barents Sea and central and southwestern (and possibly northwestern?)
Spitsbergen indicates that the northwestern, northeastern and southwestern basement terranes of
the Svalbard Archipelago were most likely already accreted together and attached to the Barents
Sea, northern Norway and northwestern Russia in the late Neoproterozoic (ca. 600 Ma). Svalbard's
three terranes were previously thought to have been juxtaposed during the Caledonian and
Ellesmerian orogenies through hundreds–thousands of kilometers of displacement along presumed
thousands of kilometers long N–S-striking strike-slip faults like the Billefjorden Fault Zone
(Harland, 1969; Harland et al. 1992, Labrousse et al., 2008; Figure 2). The presence of laterally
continuous (undisrupted), hundreds–thousands of kilometers long, Timanian thrust systems from
southwestern and central Spitsbergen to the northern Norwegian and Russian Barents Sea clearly
shows that this is not possible.
The continuous character of these thrust systems from potentially as far as onshore
northwestern Russia through the Barents Sea and Svalbard precludes any major strike-slip
displacement along N–S-striking faults such as the Billefjorden Fault Zone and Lomfjorden Fault
Zone (as proposed by Harland et al., 1974, 1992; Labrousse et al., 2008) and any hard-linked
connection between these faults in Svalbard and analogous, NE–SW-striking faults in Scotland in
the Phanerozoic (as proposed by Harland, 1969). Instead, the present work suggests that the crust
constituting the Barents Sea and the northeastern and southwestern basement terranes of Svalbard
should be included as part of Baltica in future Arctic plate tectonics reconstructions for the late
Neoproterozoic–Paleozoic period (i.e., until ca. 600 Ma). It also suggests that the Caledonian suture
zone, previously inferred to lie east of Svalbard in the Barents Sea (e.g., Gee and Teben'kov, 2004;
Breivik et al., 2005; Barrère et al., 2011; Knudsen et al., 2019) may be located west of the presently
described Timanian thrust systems, i.e., probably west of or in western Spitsbergen where





Caledonian blueschist and eclogite metamorphism was recorded in Precambrian basement rocks
(Horsfield, 1972; Dallmeyer et al., 1990a; Ohta et al., 1995; Kosminska et al., 2014).

**Conclusions**
1) Seismic data in the northern Norwegian Barents Sea and Svalbard reveal the existence of

several systems of hundreds–thousands of kilometers long, several kilometers thick, top-

SSW Timanian thrusts comprised of brittle–ductile thrusts, mylonitic shear zones and

associated SSW-verging folds that appear to extend from onshore northwestern Russia to

the northern Norwegian Barents Sea and to central and southwestern Spitsbergen. A notable

structure is the Kongsfjorden–Cowanodden fault zone in Svalbard and the Norwegian

Barents Sea, which likely merges with the Baidaratsky fault zone in the Russian Barents

Sea and southern Novaya Zemlya.

2) In the east (away from the Caledonian collision zone), these Timanian thrusts systems were

folded into NNE-plunging folds, offset, and reactivated as and/or overprinted by top-west,

oblique-slip sinistral-reverse, brittle–ductile thrusts during subsequent Caledonian (e.g.,

Agardhbukta Fault segment of the Lomfjorden Fault Zone) and, possibly, during Eurekan

contraction, and are deeply buried. By contrast, in the west (near or within the Caledonian

collision zone), Timanian thrusts were intensely deformed, pushed into sub-vertical

positions, extensively overprinted, and exhumed to the surface.

3) In eastern Spitsbergen, a major NNE-dipping Timanian thrust system, the Kongsfjorden–

Cowanodden fault zone, is crosscut by a swarm of Devonian (–Mississippian?) dykes that

intruded contemporaneous sedimentary strata deposited during extensional reactivation of

the eastern portion of the thrust system as a low-angle extensional detachment during late–

post-Caledonian collapse.

4) Timanian thrust systems were overprinted by NNE-dipping, brittle normal faults in the late

Paleozoic during the collapse of the Caledonides and/or subsequent rifting in the Devonian–

Carboniferous.

5) Timanian thrust systems and associated Caledonian and Devonian–Carboniferous brittle

overprints (e.g., Agardhbukta Fault) in the northwestern Norwegian Barents Sea and

Svalbard were mildly reactivated during the early Cenozoic Eurekan tectonic event, which

resulted in minor folding and minor reverse offsets of Devonian–Mesozoic sedimentary





strata and intrusions. Timanian thrusts and related overprints in the northeastern Norwegian
Barents Sea were not reactivated during the Eurekan tectonic event.
6) The presence of hundreds–thousands of kilometers long Timanian thrust systems suggests
that the Barents Sea and Svalbard's three basement terranes were already attached to
northern Norway and northwestern Russia in the late Neoproterozoic (ca. 600 Ma),
precludes any major strike-slip movements along major N–S-striking faults like the
Billefjorden and Lomfjorden fault zones in the Phanerozoic, and suggests that the
Caledonian suture zone is located west of or in western Spitsbergen.

**Acknowledgements**
The present study was supported by the Research Council of Norway, the Tromsø Research
Foundation, and six industry partners through the Research Centre for Arctic Petroleum
Exploration (ARCEx; grant number 228107), the SEAMSTRESS project (grant number 287865),
and the Centre for Earth Evolution and Dynamics (CEED; grant number 223272). We thank the
Norwegian Petroleum Directorate and the Federal Institute for Geosciences and Natural Resources
(BGR) for granting access and allowing publication of seismic, magnetic and gravimetric data in
Svalbard and the Norwegian Barents Sea. Prof. Steffen Bergh, Dr. Winfried Dallmann, Prof. Jiri
Konopasek, Assoc. Prof. Mélanie Forien, Dr. Kate Waghorn (UiT The Arctic University of Norway
in Tromsø), Dr. Peter Klitzke (Federal Institute for Geosciences and Natural Resources –
Germany), Anna Dichiarante (Norway Seismic Array), Rune Mattingsdal (Norwegian Petroleum
Directorate), and Prof. Carmen Gaina (Centre for Earth Evolution and Dynamics, University of
Oslo) are thanked for fruitful discussion.

**Data availability**
For high-resolution versions of the figures and supplements, the reader is referred to the
Open Access data repository DataverseNO (doi.org/10.18710/CE8RQH). The complete seismic
study is also available from the corresponding author upon request.

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



**Figures**





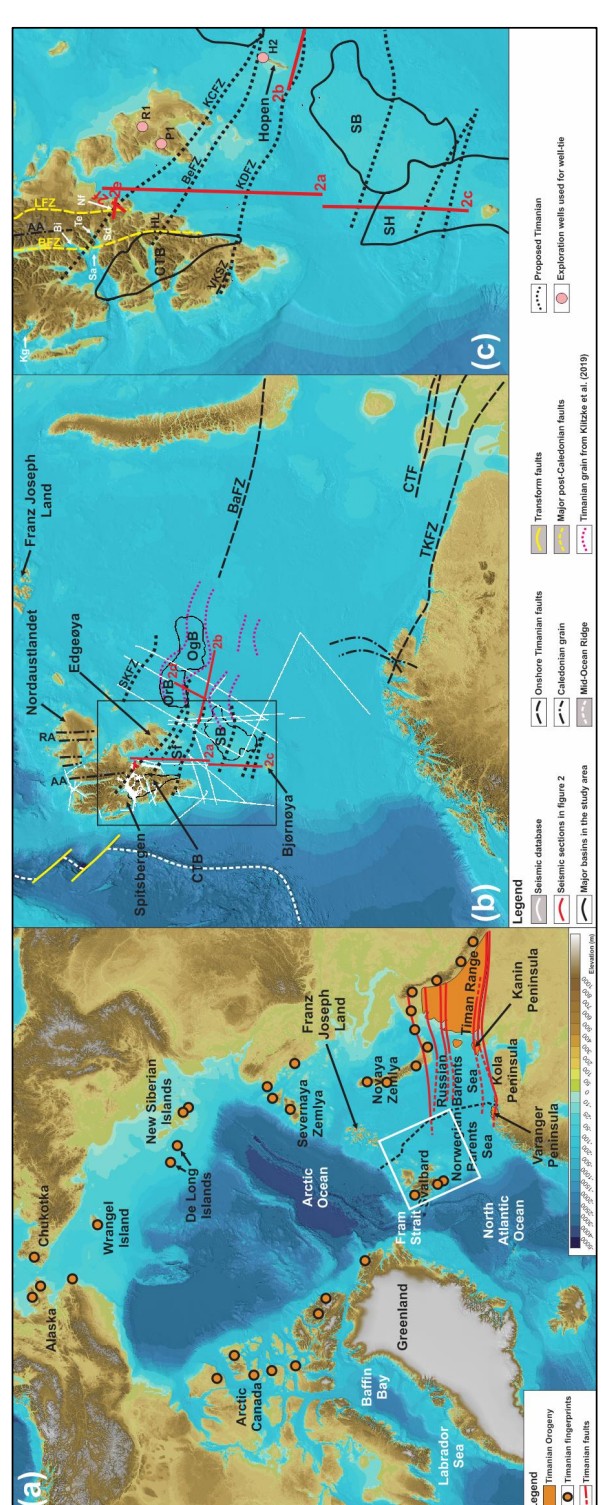



Figure 1: (a) Overview map showing the Timanian belt in Russia and Norway, and occurrences of Timanian fingerprints throughout the Arctic; (b) Regional map of Svalbard and the Barents Sea the main geological elements and the seismic database used in the present study. The location of (b) is shown as a white frame in (a); (c) Zoom in the northern Norwegian Barents Sea and Svalbard showing the main faults and basins in the study area, and the proposed Timanian structures. The location of (c) is shown as a black frame in (b). The location of the Raddedalen-1 well is from Smyrak-Sikora et al. (2020). Topography and bathymetry are from Jakobsson et al. (2012). Abbreviations: AA: Atomfjella Antiform; BaFZ: Baidaratsky fault zone; BeFZ: Bellsundbanken fault zone; BFZ: Billefjorden Fault Zone; Bi: Billefjorden; CTB: Central Tertiary Basin; HL: Heer Land; H2: Hopen-2; KCFZ: Kongsfjorden–Cowanodden fault zone; KDFZ: Kinnhøgda–Daudbjørnpynten fault zone; Kg: Kongsfjorden; LFZ: Lomfjorden Fault Zone; Nf: Nordmannsfonna; OgB: Olga Basin; OrB: Ora Basin; P1: Plurdalen-1; RA: Rijpdalen Anticline; R1: Raddedalen-1; Sa: Sassenfjorden; SB: Sørkapp Basin; Sf: Storfjorden; SH: Stappen High; SKFZ: Steiløya–Krylen fault zone; Te: Tempelfjorden; TKFZ: Trollfjorden–Komagelva Fault Zone; VKSZ: Vimsodden–Kosibapasset Shear Zone.





**Figure 2: Paleogeographic reconstruction of the Svalbard Archipelago in the latest Neoproterozoic during the Timanian Orogeny and in the early–mid Paleozoic during the Caledonian Orogeny according to previous models (e.g., Harland, 1969; Labrousse et al., 2008).**



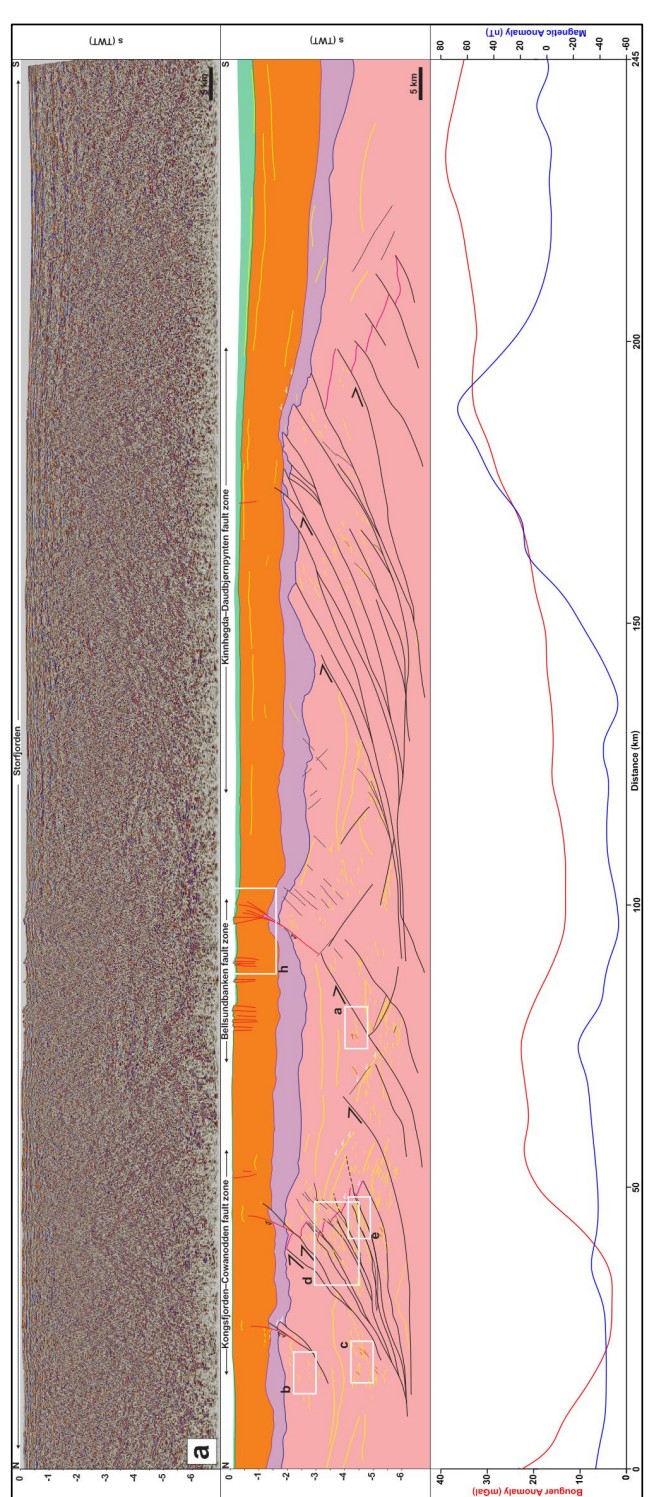

1532





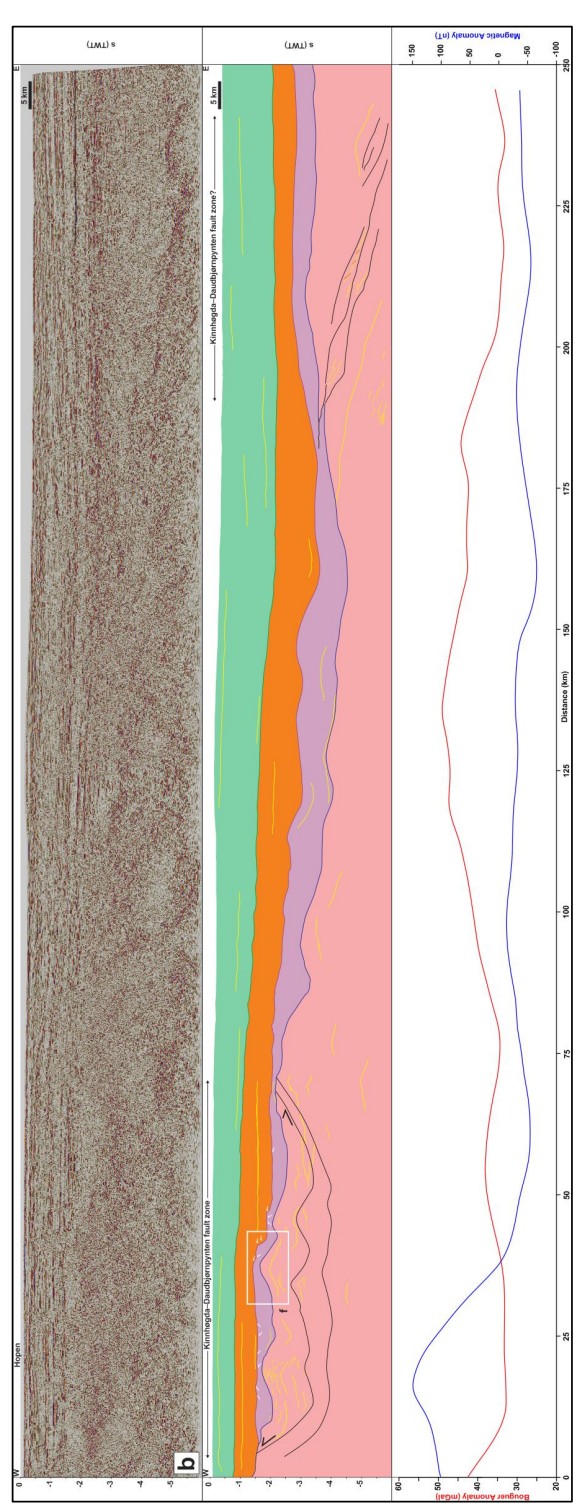

1533



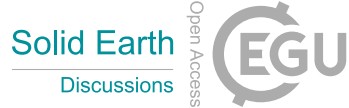

1534



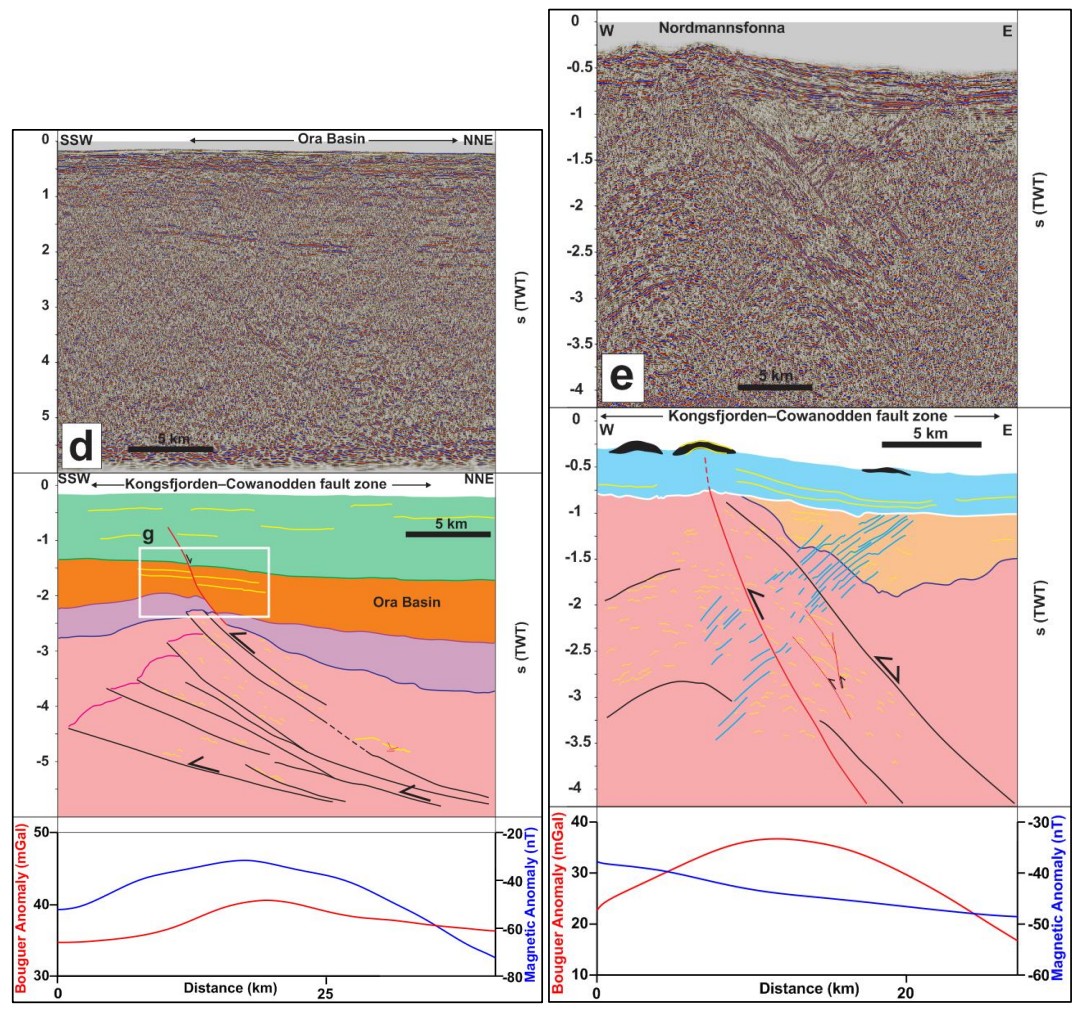

1535



Figure 3: Interpreted seismic profiles and associated potential field data (a) in Storfjorden, (b) south of Hopen, (c) on the Stappen High in the northwestern Norwegian Barents Sea between Spitsbergen and Bjørnøya, (d) on the southern flank of the Ora Basin in the northeastern Norwegian Barents Sea, and (e and f) in Nordmannsfonna in eastern Spitsbergen. The seismic profiles show top-SSW Timanian thrusts that were reactivated and overprinted during subsequent tectonic events such as Caledonian contraction, Devonian–Carboniferous late–post Caledonian collapse and rifting, Eurekan contraction, and present-day contraction. Profiles (e) and (f) also show Paleozoic and Cretaceous intrusions. The white frames show the



location of zoomed-in portions of the profiles displayed in Figure 4. Potential field data below the seismic profiles include Bouguer anomaly (red lines) and magnetic anomaly (blue lines). The potential field data show consistently high gravimetric anomalies and partial correlation with high magnetism towards the footwall of each major thrust systems (i.e., towards thickened portions of the crust).



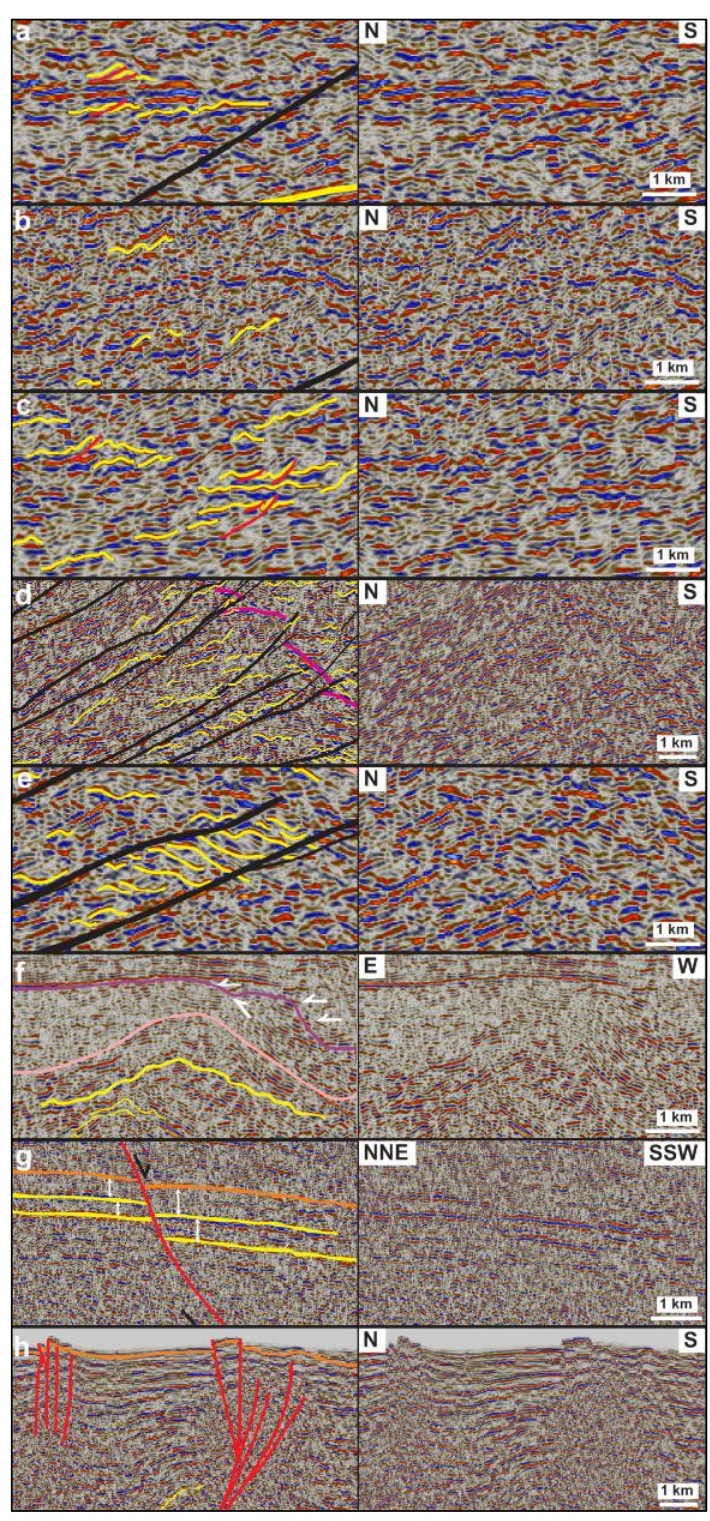





**Figure 4: Zooms in seismic profiles shown in Figure 3 showing (a) upright fold structures, (b) SSW-verging folds and (c) top-SSW minor thrusts in Precambrian–lower Paleozoic (meta-) sedimentary basement rocks, (d) SSW-verging folds and NNE-dipping mylonitic shear zones within a major thrust that offsets major basement unconformities (fuchsia lines) top-SSW, (e) duplex structures within a major top-SSW thrust, (f) a N–S- to NNE–SSW-trending, 5–15 kilometers wide, symmetrical , upright macro-fold and associated, kilometer- to hundreds of meter-scale, parasitic macro- to meso-folds, (g) syn-tectonic thickening in Devonian–Carboniferous (–Permian?) sedimentary strata offset down-NNE by a normal fault that merges with a thick mylonitic shear zone at depth, and (h) recent–ongoing reverse offsets of the seafloor reflection by multiple, inverted, NNE-dipping normal faults in Storfjorden. See Figure 3 for location of each zoom and for legend.**



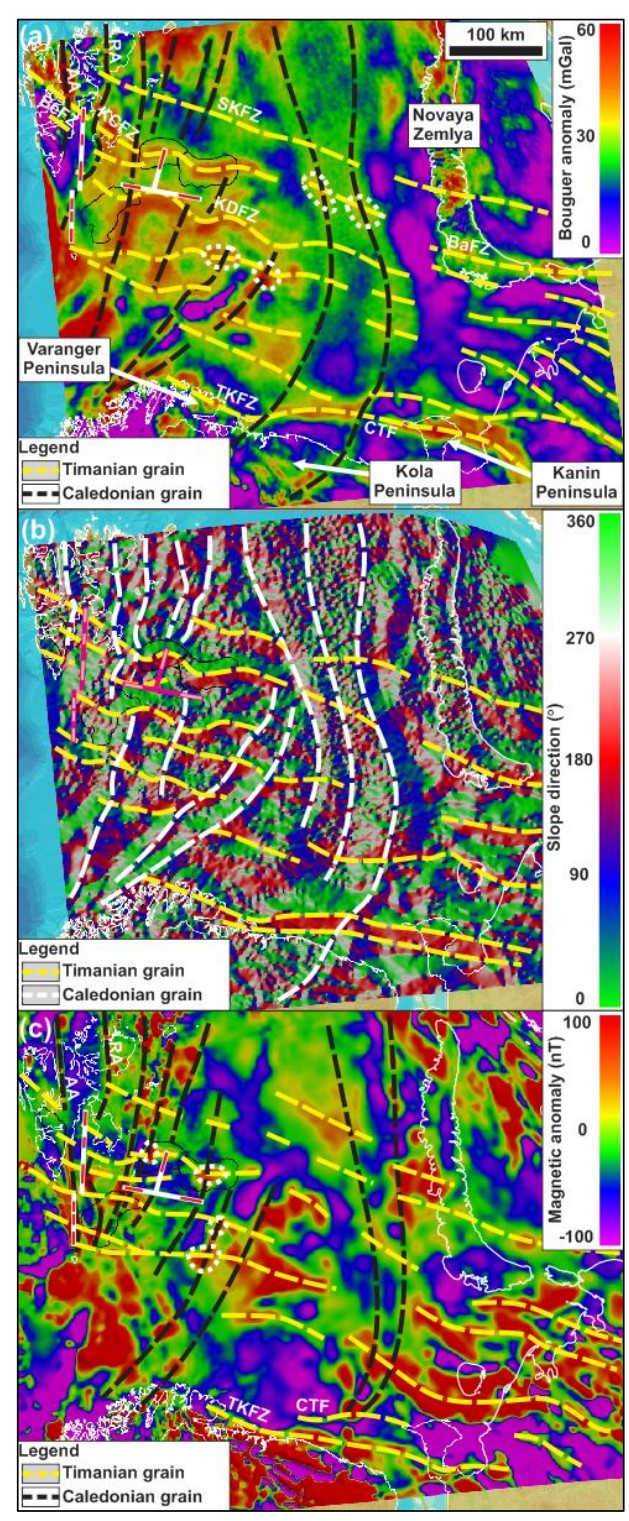





**Figure 5: Gravimetric (a and b) and magnetic (c) anomaly maps over the Barents Sea and adjacent onshore areas in Russia, Norway and Svalbard showing E–W- to NW–SE-trending anomalies (dashed yellow lines) that correlate with the proposed NNE-dipping Timanian thrust systems in Svalbard and the northern Norwegian Barents Sea. Note the high obliquity of E–W- to NW–SE-trending Timanian grain with NE–SW- to N–S-trending Caledonian grain (dashed black/white lines). Note that dashed lines in (a) and (c) denote high gravimetric and magnetic anomalies. Also notice the oval-shaped high gravimetric and magnetic anomalies (dotted white lines) at the intersection of WNW–ESE- and N–S- to NNE–SSW-trending anomalies in (a) and (c) resulting from the interaction of the two (Timanian and Caledonian) thrust and fold trends. The location of seismic profiles presented in Figure 3a–d are shown as thick white lines in (a) and (c) and as fuchsia lines in (b). Within these thick white and fuchsia lines, the location and extent of thrust systems evidenced on seismic data (Figure 3) is shown in white in (a) and (c) and in pink in (b). For the E–W-trending seismic profile shown in Figure 3b, this implies that the red and pink lines represent N–S-trending synclines. Abbreviations: AA: Atomfjella Antiform; BaFZ: Baidaratsky fault zone; BeFZ: Bellsundbanken fault zone; CTF: Central Timan Fault; KCFZ: Kongsfjorden–Cowanodden fault zone; KDFZ: Kinnhøgda–Daudbjørnpynten fault zone; RA: Rijpdalen Anticline; SKFZ: Steiløya–Krylen fault zone; TKFZ: Trollfjorden–Komagelva Fault Zone.**







**Figure 6: Sketches showing a possible reconstruction of the tectonic history of the E–W seismic profile in Nordmannsfonna shown in Figure 3e. (a) Formation of a NNE-dipping, mylonitic thrust system (Kongsfjorden–Cowanodden fault zone) within Precambrian basement rocks during the Timanian Orogeny in the latest Neoproterozoic. The NNE-dipping Kongsfjorden–Cowanodden fault zone appears near horizontal on the E–W transect; (b) Top-west thrusting along the east-dipping Agardhbukta Fault and folding of the Kongsfjorden–Cowanodden fault zone into a broad, moderately NNE-plunging anticline during the Caledonian Orogeny; (c) Inversion of the Kongsfjorden–Cowanodden fault zone along the eastern flank of the Caledonian anticline and deposition of thickened, gently west-dipping, syn-tectonic, Devonian (–Mississippian?) sedimentary strata during post-Caledonian collapse-related extension; (d) Intrusion of Precambrian basement and Devonian (–Mississippian?) sedimentary rocks by steeply west-dipping dykes in the Devonian–Mississippian; (e) Regional erosion in the mid-Carboniferous (latest Mississippian) and deposition of Pennsylvanian sedimentary strata, possibly along a high-angle brittle splay of the inverted portion of the Kongsfjorden–Cowanodden fault zone during rift-related extension; (f) Deposition of Mesozoic sedimentary strata and intrusion of Cretaceous dolerite dykes and sills; (g) Erosion of Pennsylvanian–Mesozoic strata and reactivation of the Kongsfjorden–Cowanodden fault zone and Agardhbukta Fault with minor reverse movements in the early Cenozoic during the Eurekan tectonic event as shown by mild folding and offset of overlying post-Caledonian sedimentary strata, dykes and Base-Pennsylvanian unconformity. Also note the back-tilting (i.e., clockwise rotation) of Devonian–Mississippian dykes in the hanging wall of the Agardhbukta Fault and of the Kongsfjorden–Cowanodden fault zone.**





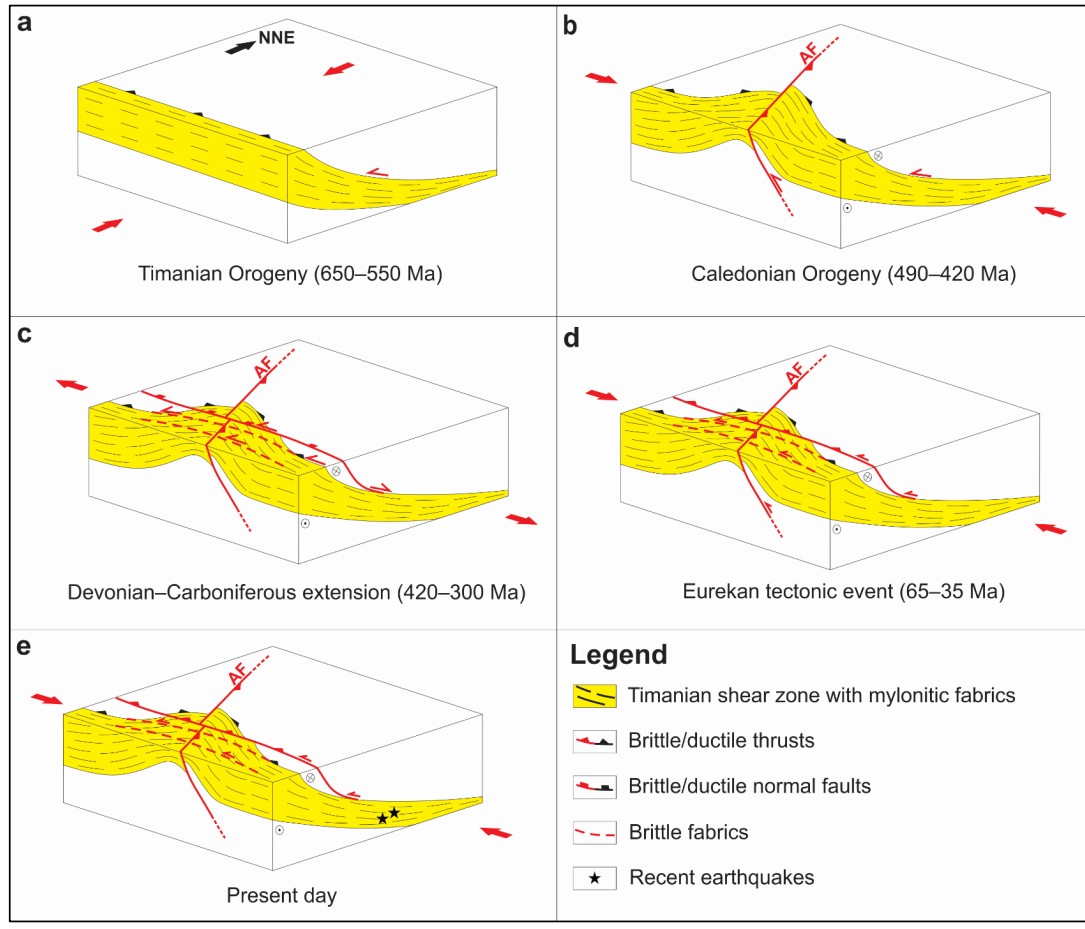

**Figure 7: Tectonic evolution of Timanian thrust systems in eastern Spitsbergen, Storfjorden and the northwestern Norwegian Barents Sea including (a) top-SSW thrusting during the Timanian Orogeny, (b) reactivation as oblique-slip sinistral-reverse thrusts and offset by top-west brittle thrust overprints (e.g., Agardhbukta Fault – AF) under E–W contraction during the Caledonian Orogeny, (c) reactivation as low-angle, brittle–ductile, normal–sinistral extensional detachments and overprinting by high-angle normal–sinistral brittle faults during Devonian–Carboniferous, late–post-Caledonian extensional collapse and rifting, (d) reactivation as brittle–ductile sinistral–reverse thrusts, overprinting by high-angle sinistral–reverse brittle thrusts, and mild offset by reactivated top-west thrusts (e.g., Agardhbukta Fault – AF) during E–W Eurekan contraction, and (e) renewed, recent–ongoing, sinistral–reverse reactivation and overprinting possibly due to ongoing magma extrusion and transform faulting (ridge-push?) in the Fram Strait.**