# Peer review of "Impact of Timanian thrust systems on the late"

_Solid Earth, 2021_

## Author Response (AR1)

**Reply to Anthony Doré**

Dear Prof. Doré, thank you very much for your input on the manuscript, it is highly appreciated. Here is our reply to your comments. We hope the changes we implemented improve the shortcomings of the manuscript highlighted by your comments and suggestions. Please do not hesitate to contact us shall this not be the case for some comments.

**1. Comments from Prof. Doré**

Comment 1: This very interesting paper should certainly be published in Solid Earth, not because it provides a definitive solution to the Barents Sea tectonic mosaic, but because it provides a bold and well-argued alternative to the current basement models. In essence, the model would glue the pieces of the Barents Sea (and specifically Svalbard) together in the Neoproterozoic, thus eliminating the idea of later Caledonian assembly from different terranes. It would also put paid to the concept of "Barentsia", the putative microcontinent including Svalbard, formed between two supposed sutures representing arms of the Caledonian orogen.

Comment 2: So how convincing is the interpretation of the Timanian grain extending across from Russia to Svalbard? The idea has to some extent been "out there" for a while, based on the distinct cross-cutting trend of the Tiddlybanken Basin, and the offshore extension of the Trollfjord-Komagelv fault (not strictly Timanian, and actually contested by the lead author elsewhere). But my first reaction on seeing the seismic lines in Fig. 3 was scepticism that such detailed interpretations of deep basement structure could be made from such difficult data. Having to look at the seismic lines with my neck at a right-angle didn't help..... Yes I can see what might be thrust stacks, but I can see lots of other patterns too. That's what happens when you look at noisy data. However, it's good that this is an open review. Others can also take a look and tell me whether I'm being fair or not. Also in the interests of being fair, the authors have looked at a lot more seismic than I have, and for longer - plus the local panels shown in Figure 4 are more convincing.

Comment 3: A significant weakness in the paper's argument, acknowledged by the authors, is the non-observation of these Timanian structures on Svalbard despite the existence of a few apparently Timanian metamorphic dates. You would really think such a pervasive and dominant tectonic regime, actually not very deeply buried on the seismic lines, would be expressed onshore. The reasons given for the absence are depth of burial and obscuring by later superimposed tectonic events. The latter argument, in particular, is pretty thin based on what the seismic leads us to expect.

Comment 4: Despite these reservations, i think this one can get by with minor revision only. It is very well-written (thanks!) and the illustrations are mainly of good quality. I suggest the following improvement:

1) The multi-panel figures (particularly 3 and 4) are confusing, particularly in online formal. Why not simply call all the different panels different figures? It would make things so much easier.

Comment 5: 2) In any case, the seismic sections are wrongly labelled on Fig. 1. Unless I've got it badly wrong, they should read 3a, 3b etc., (not "2"). The first three sections need readable direction indicators (N-S etc.).

Comment 6: 3) The paper badly needs its own plate tectonic reconstruction (not just the one that is being challenged, Fig. 2). What is your alternative? How was the Barents basement assembled?

Comment 7: Pushing the Iapetus suture even farther west is OK, but where specifically?

Comment 8: And what is the implication for Caledonian assembly, and supposed Laurentian affinity of parts of Svalbard?

Comment 9: How does this idea fit with Caledonian thrust sheets extending as far east as (south of) the Varanger Peninsula? A few simple sketches would suffice to show what you are thinking.

Comment 10: Although these are comparatively minor changes, i look forward to seeing what others say. I'm happy to take a quick look at any revisions. I'm still not sure whether I agree with the idea, but I strongly believe it should be out there and part of the debate.

**2. Author's reply**

Comment 1: agreed.

Comment 2: agreed, though the data the authors investigated for the present study did not include much noise. The authors of the present manuscript emphasize that high-resolution versions of the figures are available on DataverseNO to evaluate the work presented (https://dataverse.no/dataset.xhtml?persistentId=doi:10.18710/CE8RQH).

Comment 3: Seismic sections in Figure 3a–c show Timanian portions of the thrust systems (i.e., not related to Caledonian and post-Caledonian brittle overprints, e.g., listric brittle faults and offsets of the seafloor) are buried under at least 2.0–2.5 seconds (TWT; i.e., several kilometers thick) successions of Phanerozoic sediments in the Barents Sea and Storfjorden. In its shallowest segment in Nordmannsfonna where it is folded into an anticline, the Kongsfjorden–Cowanodden fault zone is still buried under at least 1.0 second (TWT) of sediments (Figure 3d–e), which still corresponds to a depth of at least 2.5–4.5 kilometers based on seismic velocities for Pennsylvanian to Cretaceous sediments from Gernigon et al. (2018). The authors of the present manuscript do not argue that Timanian fault systems are buried everywhere in the Svalbard Archipelago, especially because they were uplifted and exhumed in western Spitsbergen due to strong Caledonian contraction (and subsequent Eurekan contraction). Based on the arguments presented by previous workers for the Vimsodden–Kosibapasset Shear Zone in southern Spitsbergen (e.g., Bjørnerud, 1990; Bjørnerud et al. 1991; Majka et al., 2008, 2012; Mazur et al., 2009), it is highly probable that this major fault zone formed during the Timanian Orogeny as a top-SSW thrust, thus generating the observed regional unconformity between Neoproterozoic and latest Neoproterozoic metasedimentary rocks in the area. In addition, recent dating by Faehnrich et al. (2020) along the Vimsodden–Kosibapasset Shear Zone and other related minor fault zones in southern Spitsbergen further illustrate the point of the authors of the present manuscript about the overprinted character of Timanian thrusts in Spitsbergen. The Vimsodden–Kosibapasset Shear Zone yielded exclusively Caledonian ages (their sample 16-62A), whereas related parallel minor shear zones were only mildly reactivated–overprinted by later Caledonian contraction and preserved partly their Timanian signal (their samples 16-25A and 16-73A). This is discussed in the present manuscript lines 878–887.

Comment 4: the authors of the present manuscript understand the reviewer's perspective here, but changing the labels of Figure 3a–e may not be ideal. At the moment, when reading the manuscript, the reader will only need to remember number "3" to know that the referred "Figure 3" (a–e) refers to seismic data (and will therefore most likely not have to interrupt his/her reading to double check). However, if these labels were to be changed into Figures 3 to 7, the reader will have to remember 5 numbers and may have to disrupt her/his reading more often.

Placing the seismic lines together also aids comparison. Setting Figure 3a–e as separate figures will likely mean that they appear on different pages in the paper, meaning that readers will have to keep flipping back and forth to compare observations from different lines. As such, the authors of the present manuscript would therefore prefer to keep Figure 3a–e together. Nonetheless, the authors of the present manuscript are open to changing the labels and therefore await further instructions from both referees and from the editor.

Comment 5: agreed. However, the labels of the seismic sections are as large as they can be and, should it be necessary, they are possible to read better from the high-resolution of the figures on DataverseNO (doi.org/10.18710/CE8RQH).

Comment 6: agreed.

Comment 7: agreed. Based on the present manuscript's findings, the only natural location for the Iapetus suture is in western Spitsbergen where blueschist and eclogite of Caledonian age are recorded (Horsfield, 1972; Dallmeyer et al., 1990a; Ohta et al., 1995). This is discussed lines 1048–1053 in the present manuscript.

Comment 8: agreed. The present findings have implications both for Caledonian assembly and for the affinity of northeastern Spitsbergen with Greenland. The present manuscript does briefly discuss the tectonic implications of the presented findings in the final sub-section of the discussion. However, in order to keep the manuscript focused and to a reasonable length, these issues will be discussed in a future short manuscript, which will also integrate further datasets (e.g., paleontology) to infer a geodynamic evolution. Points of emphasis will include the use of paleontology to infer terrane separation in plate tectonics reconstructions (e.g., thousands of kilometers separation of northeastern Svalbard from Baltica based on differences in Ordovician trilobites; Fortey and Cocks, 2003).

Comment 9: agreed, although the authors of the present manuscript think that this is beyond the scope of the present study, which focuses on advancing the idea that Timanian thrust systems are present across Svalbard and the Barents Sea. Several papers are currently under development that consider the geodynamics implications of the presented findings more broadly, including one in which implications for Caledonian tectonics are considered. For examples, the idea of deep Timanian thrust systems crosscutting the whole Barents Sea and Svalbard fit well with the presence of Caledonian thrust sheets as far as southwest of the Varanger Peninsula since these represent the shallow portion of the crust. If Timanian thrusts are present at depth > 2–3 kilometers in Svalbard and the northwestern Barents Sea, it is therefore conceivable that such thrust systems are present at depth in northern Norway too. However, the commonly accepted Timanian front being the Trollfjorden–Komagelva Fault Zone on the Varanger Peninsula, it is therefore not required to discuss the impact of the present manuscript's findings on Caledonian nappes south of the

Trollfjorden–Komagelva Fault Zone and of this fault's western continuation. A key element in the upcoming years will be to further constrain the nature and location of its extent offshore onto the Finnmark Platform. The lead author has attempted to address this issue during his Ph.D. (Koehl et al., 2018, 2019). It is the lead author's belief that the models presented in both Ph.D. manuscripts are (partly to completely) wrong and need updating, especially in the light of the geometry and attitude of Timanian thrust systems in Svalbard and the Barents Sea.

Comment 10: agreed.

**3. Changes implemented**

Comment 1: none required by the reviewer.

Comment 2: none required by the reviewer.

Comment 3: none.

Comment 4: awaiting further instructions from the editor and reviewers.

Comment 5: changed the labels of the seismic sections in Figure 1 to "3a–e".

Comment 6: added the proposed plate tectonics alternative as Figure 8.

Comment 7: none.

Comment 8: none.

Comment 9: see reply to comment 6.

Comment 10: none required by the reviewer.

**4. Additional changes implemented**

-Lines 80–81: added ", and imply that the Norwegian Barents Sea and Svalbard basement may contain Timanian structures overprinted during later (e.g., Caledonian) deformation events" for clarity.

-Lines 94–100: split the sentence into two and partly rewrote it to make it easier to read.

-Line 103: added "by future research" for clarity.

-Lines 1004–1005: split the sentence into two and added ". If correct, a Timanian origin for these structures would" to make it more readable.

-Lines 1055 and 1064: added reference to the new Figure 8 as a consequence to Prof. Doré comment 6.

-Lines 1079–1080: added "We interpret these thrust systems as being related to the Neoproterozoic Timanian Orogeny." for clarity.

**Reply to Jeremy Rimando**

Dear Dr. Rimando, thank you very much for your input on the manuscript, it is highly appreciated. Here is our reply to your comments. We hope the changes we implemented improve the shortcomings of the manuscript highlighted by your comments and suggestions. Please do not hesitate to contact us shall this not be the case for some comments.

**1. Comments from Dr. Rimando**

Comment 1: I found this paper very thought-provoking. They propose an alternative mechanism and timing for the accretion of basement terranes in Svalbard and the Barents Sea. They propose that these basement terranes were accreted by top-to-the SSW thrusts faults during the Neoproterozoic 'Timanian Orogeny,' rather than by displacement along N–S-striking strike-slip faults during the Paleozoic Caledonian Orogeny. This paper really demonstrates the authors' breadth of knowledge of the previous work on the structures which they suggest belong to "continuous (undisrupted), hundreds–thousands of kilometers long, Timanian thrust systems."

Comment 2: However, I think that the paper will require a bit more work to convince readers of the presence of a laterally continuous system of Timanian thrust faults throughout Spitsbergen, Storfjorden, and the Norwegian Barents Sea. As it is, I am not convinced that the authors' interpretations of a few seismic profiles, including correlation of these interpreted structures with lineaments on gravity, magnetic, and slope direction maps, comprise compelling evidence for the lateral continuity of these WNW-ESE-striking and NNE-dipping Timanian thrusts. Ideally, they should have inspected multiple perpendicular seismic profiles from west to east and correlated these. It might help to include additional representative seismic profiles at different longitudes (and incorporate these in the supplementary file at the very least) to bolster their argument for a continuous thrust system.

Comment 3: The lineaments in the gravity and magnetic anomaly maps that the authors claim to be the continuation of the thrust faults could be anything. Even if these were faults, these might display different fault styles, kinematics, and/or timing of deformation. Granted that observing other kinematics on these WNW-ESE-striking faults does not rule out the possibility that these are the prolongation of the Timanian thrusts (i.e., overprinting may have happened), interpreting more seismic profiles and including a discussion similar to the section 'Devonian–Carboniferous normal overprint–reactivation' should help clarify this.

Comment 4: In short, I do not think the spatial coverage of the data and the amount of analysis conducted is sufficient to suggest the presence of such a large, continuous thrust system. The authors could either do additional analyses or at least describe their level of confidence in their mapping of different portions of the fault system, and be clear about which traces are speculative and which traces are certain.

Comment 5: In some instances, it's difficult to follow their line of reasoning for describing a lateral continuous Timanian thrust fault system. They claim that the structures they observed in the northwestern Norwegian Barents Sea are comparable to structures observed onshore and offshore in other areas, but it is unclear how some of their descriptions support such claims. For instance, the Vimsodden–Kosibapasset Shear Zone (VKSZ) is dominantly strike-slip. It is not clear how this is proof that the VKSZ displays similar configuration and kinematics and, consequently, represents the westward continuation of the Kinnhøgda–Daudbjørnpynten fault zone.

Comment 6: They describe associated map view folds and they explain the VKSZ's strike-slip kinematics, albeit much later in the text, through strong overprinting by the Caledonian Orogeny. What is the scale and timing of the folds that are observed in map view along the VKSZ? Is there proof that these are Timanian and not folding related to the Caledonian ductile shear zone? While later paragraphs seem to clarify the nature of this folding, the manner in which the VKSZ example is presented as proof does not seem convincing. Instead, it creates confusion. I only cite one example, but I suggest that the authors review how they presented their other arguments for an extensive Timanian thrust system.

Comment 7: As noted by Tony Doré (Reviewer 1), I am also not convinced with why these major thrust systems in Svalbard went unnoticed before. They argue that strong overprinting of the VKSZ by Caledonian Orogeny explains why such thrust systems were not identified before, but in an earlier paragraph they describe folds in map view (which are presumably large and obvious) as proof of the onshore continuation of this thrust system. I would expect to see more exposures of the Timanian thrusts onshore, despite 'deep burial' since, as they themselves claim, these areas onshore would have been intensely deformed, and most likely experienced high uplift and exhumation due to their proximity to the Caledonian collision zones.

Comment 8: On its own, this paper doesn't really provide definitive evidence of the presence of hundreds-thousands of kilometers long Timanian thrust systems and I think this issue should be addressed before they even consider exploring the impact of the existence of Timanian thrust systems on the tectonic evolution of the region.

Comment 9: Besides, considering that they discuss the impact of these thrust systems on the tectonic evolution of the region, the authors should include schematic diagrams, or better yet, time-lapse images of their proposed plate reconstruction model.

Comment 10: Overall, the paper is well written. A few stylistic changes, including tweaks to figures and consistency in using in-text citation of figures and figure labels, will significantly improve the readability of this paper. Below are a few minor technical comments to consider:

1)    Please make sure that all features/places (e.g., Baltica, Caledonides, Norway, Laurentia, Pearya, Sassendalen, Hornsund) you described are included in your maps. In all of the sections, figures (and panel letters) should be cited consistently in the text right after the feature being described to make it easier for readers who are not familiar with the area to locate the features you are referring to. Please also make sure that the labels on the maps are big enough and easy to read. Some of the text might need to be outlined in another color to provide a contrast to the background and some may have to be brought to the topmost layer items on your figure to prevent them from being blocked by other lines/shapes.

Comment 11: 2)    I suggest indicating the ages of these 'Timanian fingerprints' on the map to emphasize the contemporaneity of structures and citing the corresponding references on the figure captions as well.

Comment 12: 3)    Indicate the abbreviations of geologic features and places in the text, similar to how you did in the figures (e.g., BAFZ for the Baidaratsky fault zone), so that it is easier to locate them on the maps.

Comment 13: 4)    Please include a north arrow, a scale bar, and northing and easting labels around the map frame. It's difficult to visualize some descriptions of fault lengths in the text since you did not put any scale bars on your map in fig 1.

Comment 14: 5) The authors plot the other seismic profiles that belong to the DISKOS database on a map, which is good, but these should be labeled and cited in the text alongside citations of previous studies that inspected these particular seismic lines. If there are other previous studies that look into seismic profiles that are not part of the DISKOS database, these should be included as well. The locations of previous studies which were discussed to provide evidence of the lateral extent of these Timanian thrusts should also be plotted.

Comment 15: 6)   Rippington et al. (2010), and the lead author himself in Koehl (2018), cast doubt on the existence of an 'Ellesmerian Orogeny' due to the lack of compelling evidence from cross-cutting relationships and age constraints, but 'Ellesmerian Orogeny' is mentioned several times in the text.

Comment 16: 7)   Is 'top-SSW', 'top-E', or 'top-S' standard notation? Why not use 'top-to-SSW'/ 'top-to-the-SSW', 'top-to-east'/'top-to-the-east', or 'top-to-south'/ 'top-to-the-south' instead?

Comment 17: 8)   I think it is necessary to outline the approximate extent of the Precambrian basement terranes on a map.

Comment 18: 9)   In the section geologic setting, can you describe the orientation of the structures (e.g., N-S-striking BFZ and LFZ) as well as the direction of the maximum horizontal stress (and changes thereof) associated with each major tectonic event, to provide context for the kinematics of the structures you describe?

Comment 19: 10)   Is there a specific reason for using 'interpret basement-seated structures' instead of 'basement-structures?' It seems like a combination of 'basement-structures' and 'deep-seated.'

Comment 20: 11)   Double check the labelling of figures, especially of the seismic profiles on the map (figure 1).

Comment 21: 12)   In figure 2, what do you mean by main tectonic stress? Do you mean direction of maximum horizontal stress?

Comment 22: 13)   I don't think yellow is the best color to outline reflectors in the pink and purple units in your seismic profile interpretations.

Comment 23: 14)   Indicate the location of the potential field data in figure 5 on the map (figure 1) using a box.

Comment 24: 15)   2D seismic profiles only give you the vertical component of displacement, and don't really give a complete picture of the kinematics of faulting.  I wonder if the faults you describe as thrust could be oblique or dominantly strike-slip?

Comment 25: 16)   The authors cite the paper Koehl et al (in review) a lot. Please refer to the guidelines of EGU (Copernicus Publications) on citations of unpublished work.

Comment 26: 17)    Check completeness/accuracy of descriptions of different figure panels and features on figures. Figure 5b shows a slope direction map, but the caption says it's a gravity map.

Comment 27: 18)    The authors write in the passive voice too much. I think it's fine to write in the active voice to avoid making sentences too wordy and difficult to understand.

Comment 28: 19)    Please make sure if saying "The complete seismic study is also available from the corresponding author upon request" complies with Copernicus Publications' commitment to the 'Coalition on Publishing Data in the Earth and Space Sciences' (COPDESS) and the 'Enabling FAIR (findability, accessibility, interoperability, and reusability) Data Commitment Statement in the Earth, Space, and Environmental Sciences.'

Comment 29: It was a pleasure reviewing your interesting work! I believe the paper is worthy of being published in Solid Earth after addressing the issues I raised. I look forward to hearing your thoughts and I'd be happy to a look at a revised version of this manuscript.

**2.  Author's reply**

Comment 1: agreed.

Comment 2: agreed. However, if Dr. Rimando is referring to the lack of seismic data in the Russian Barents Sea, it is not possible to obtain data on Russian territory outside of Russia and one must physically go to Russia to interpret such data. Thus, for mapping of the Baidaratsky fault zone, the authors of the present manuscript rely on previous seismic interpretation and onshore–offshore by Prof. Lopatin and Prof. Korago (Lopatin et al., 2001; Korago et al., 2004), which are summaries of mapping campaigns in the Russian Barents Sea and onshore northwestern Russia. Regarding the Norwegian sector of the Barents Sea, the authors of the present manuscript did look at many more N-S and E-W profiles but had not secured permission to show these prior to submitting the manuscript. Figure 1 attached the present response to Dr. Rimando's comments shows the whole seismic database used for the present study. The authors of the present manuscript have now secured permission to show the whole dataset and, in addition to those presented in Figure 3a–e or in the supplementary data, the authors of the present manuscript direct the reader to the DISKOS database.

Comment 3: agreed. The lineaments in the gravity and magnetic anomaly map could indeed be anything and the present manuscript does not have the ambition of providing a definitive answer to this. However, the present manuscript presents evidences suggesting that they may represent

Timanian faults and/or folds. Timanian faults–folds onshore Russia with the exact same WNW–ESE strike/trend as those mapped on seismic data in the Norwegian Barents Sea and Svalbard correlate with the eastern continuations of the gravimetric and magnetic anomalies Timanian thrusts (and folds) that coincide with Timanian thrusts in the Norwegian Barents Sea and Svalbard. This interpretation is also backed up by previous studies in Russia (Lopatin et al., 2001; Korago et al., 2004), which have successfully mapped the largest of these Timanian faults all the way to the border with the Norwegian Barents Sea, where they coincide with the eastern continuation of the Kongsfjorden–Cowanodden fault zone (Baidaratsky fault zone) and Trollfjorden–Komagelva Fault Zone (Central Timan Fault; see Figures 1 and 5). The fact that these anomalies display relatively homogeneous character from the Norwegian Barents Sea to the Russian Barents Sea and onshore northwestern Russia further suggest that the geometries and kinematics (and, quite possibly, the timing of formation) of the faults (and folds) are consistent throughout these areas, thus further supporting the model proposed. The only exception would be towards the Uralides farther east in Russia, and in central–western Spitsbergen where these faults would have been strongly overprinted (e.g., Vimsodden–Kosibapasset Shear Zone reactivated as a sinistral strike-slip fault in Caledonian times and rotated into a subvertical fault – Faehnrich et al., 2020; Kongsfjorden–Cowanodden fault zone folded into north- to NNE-plunging Caledonian folds and overprinted by Devonian–Carboniferous brittle normal faults, which were themselves inverted in Cenozoic times in Svalbard and Storfjorden, i.e., close to the active Cenozoic margin of western Spitsbergen, but not farther east). These exceptions are discussed in the present manuscript in section "Phanerozoic reactivation and overprinting of Timanian thrust systems" (starting line 810).

Comment 4: agreed. The authors of the present manuscripts used more seismic data than is available as figures in the manuscript (see response to comment 2) and therefore do believe that spatial coverage of the data is sufficient to support their argumentation (see attached Figure 1 showing the complete seismic database). The authors of the present manuscript also note that they tried to use language throughout the manuscript that acknowledges that their interpretations are tentative. As both reviewers agree, the aim of this work is not to promote a definitive idea but to offer interpretations and a conceptual model that needs to be further tested. However, the authors of the present manuscript concede that the more speculative portions of the mapped faults should be highlighted in Figure 1.

Comment 5: agreed. The Vimsodden–Kosibapasset Shear Zone does display indications for sinistral strike-slip movements. However, these were recently dated to be Caledonian in age (Faehnrich et al., 2020; their sample 16-62A), thus attesting of the reactivation–overprinting history of Timanian faults during subsequent events. Nonetheless, it is clear that amphibolite facies metamorphism along the Vimsodden–Kosibapasset Shear Zone was coeval with the formation of a regional latest Neoproterozoic unconformity north of the shear zone in southwestern Spitsbergen (Bjørnerud, 1990; Bjørnerud et al., 1991; Majka et al., 2008, 2012; Mazur et al., 2009), i.e., similar to the configuration and deformation intensity along Timanian thrust systems in the Barents Sea and Svalbard. Considering the obliquity of the (most likely) Timanian Vimsodden–Kosibapasset Shear Zone to subsequent E–W Caledonian contraction, the WNW–ESE-striking shear zone would have been ideally oriented to be reactivated as a sinistral strike-slip fault in Caledonian times (e.g., Figure 7b in the present manuscript).

Comment 6: map-view folding along all Timanian thrust systems in Svalbard and the Barents Sea are inferred to be Caledonian in age, not Timanian. The Timanian Orogeny is believed to have been a relatively simple event in the Barents Sea and Svalbard's crust, involving top-SSW thrusting along a series of dominantly NNE-dipping thrust systems. Later on, these thrust systems which were oriented highly oblique to subsequent E–W Caledonian contraction) were reactivated as sinistral strike-slip faults (e.g., Vimsodden–Kosibapasset Shear Zone; Mazur et al., 2009; Faehnrich et al., 2020) and folded into N–S-trending (north-plunging; due to the north-northeastwards dip of the thrusts) folds. These map-view N–S-trending, NNE-plunging folds (see illustration of the geometry of the folds in Figure 3d in E–W cross section and Figure 3e in N–S along-strike section) are not directly related to sinistral strikes-slip reactivation of Timanian faults but represent more gentle deformation of the thrust systems away from the Caledonian margin in western Spitsbergen. Figure 7b illustrates how WNW–ESE-striking Timanian faults were reactivated as sinistral strike-slip faults and/or folded into N–S-trending, NNE-plunging folds during the Caledonian Orogeny. However, the authors of the present manuscript concede that the figure does not illustrate the variation in the intensity of Caledonian reactivation–overprinting along Timanian faults. Timanian faults were intensely deformed along the Caledonian margin in western Spitsbergen and reactivated as sinistral strike-slip faults and folded (e.g., Vimsodden–Kosibapasset Shear Zone), whereas they were only folded in the Barents Sea and eastern Spitsbergen away from the Caledonian margin (Figure 3b). The authors of the present manuscript are open to redesign/update Figure 7 to include such along-strike variations in reactivation–overprinting intensity should it be judged necessary by both referees and the editor. These along-strike variations also apply to post-Caledonian deformation as shown by the contrast between Figure 3a and Figure 3c where the Kongsfjorden–Cowanodden fault zone was overprinted by Devonian–Carboniferous listric normal faults that were later inverted due to Eurekan contraction in Storfjorden (Figure 3a) and Svalbard (Koehl, 2021 and supplement S2c) but were not inverted farther east, away from the West Spitsbergen Fold-and-Thrust Belt margin (Figure 3c).

Comment 7: disagreed. Timanian thrusts systems onshore Svalbard are either deeply buried and/or intensely overprinted along the western Spitsbergen margin, which was the locus of both Caledonian and Eurekan (and possibly Ellesmerian) contraction. Other arguments as to why they went unnoticed in northwestern Spitsbergen (where some Timanian ages were recorded for eclogite facies metamorphism) are (1) the remoteness of the area and the large amounts of funding required to access potential outcrops and further date them, and (2) the strongly eroded character of outcrops in glaciated areas like Svalbard. Seismic sections in Figure 3a–c clearly show that the Timanian portions of the thrust systems (i.e., not related to Caledonian and post-Caledonian brittle overprints, e.g., listric brittle faults and offsets of the seafloor) are buried under at least 2.0–2.5 seconds (TWT; i.e., several kilometers thick) successions of Phanerozoic sediments in the Barents Sea and Storfjorden. In its shallowest segment in Nordmannsfonna where it is folded into an anticline, the Kongsfjorden–Cowanodden fault zone is still buried under at least 1.0 second (TWT) of sediments (Figure 3d–e), which still corresponds to a depth of at least 2.5–4.5 kilometers based on seismic velocities for Pennsylvanian to Cretaceous sediments from Gernigon et al. (2018). Nevertheless, the authors of the present manuscript do not argue that Timanian fault systems are buried everywhere in the Svalbard Archipelago, especially because they were uplifted and exhumed in western Spitsbergen due to strong Caledonian contraction (and subsequent Eurekan contraction). Based on the arguments presented by previous workers for the Vimsodden–Kosibapasset Shear Zone in southern Spitsbergen (e.g., Bjørnerud, 1990; Bjørnerud et al. 1991; Majka et al., 2008, 2012; Mazur et al., 2009), it is highly probable that this major fault zone formed during the Timanian Orogeny as a top-SSW thrust, thus generating the observed regional unconformity between Neoproterozoic and latest Neoproterozoic metasedimentary rocks in the area. In addition, recent dating by Faehnrich et al. (2020) along the Vimsodden–Kosibapasset Shear Zone and other related minor fault zones in southern Spitsbergen further illustrate the authors' point about the overprinted character of Timanian thrusts in western Spitsbergen. The Vimsodden–Kosibapasset Shear Zone yielded exclusively Caledonian ages (their sample 16-62A), whereas related parallel minor shear zones were only mildly reactivated–overprinted by later Caledonian contraction and preserved partly their Timanian signal (their samples 16-25A and 16-73A). This is discussed in the present manuscript lines 878–887.

To the comment as to why these major thrust systems went unnoticed on seismic data despite the data have been acquired in the 90s, the issue is simple. Only very few researchers (if any at all) in the world would have known what to make out of the seismic expression of these faults. The seismic expression of (mylonitic) shear zones on seismic data was first investigated by avant-garde work by Fountain et al. (1984), Hurich et al. (1985), and a few others. But even back then, the shear zone geometries correlated to seismic signals were relatively simple and consisted of linear single mylonitic detachment surfaces. It is only recently that this research front was pushed further by innovative new works like Phillips et al. (2016) and Fazlikhani et al. (2017; to cite only a few) and that kilometers thick shear zones were eventually correlated from onshore field geometries to offshore seismic geometries. This field is being further developed here, especially considering the amount of details (down to 100 meters scale) possible to observe within Timanian thrusts systems in Storfjorden (e.g., SSW-verging asymmetric folds versus mylonitic brittle–ductile shears and detachments; see high-resolution version of Figure 3a and associated zooms in Figure 4d and e). It should also be noted that seismic data around Svalbard have mostly been investigated with emphasis on shallow Paleozoic–Cenozoic sedimentary successions in the perspective of hydrocarbon exploration and carbon storage. Deep basement structures were therefore not a priority but are now being increasingly studied (e.g., Klitzke et al., 2019).

Comment 8: partly agreed. It is true that the present manuscript does not constitute a definitive answer to the structure of basement units in the Barents Sea and Svalbard. Much further work is needed to further investigate the thrust systems described herein and to better constrain their geometry in 3D. However, it is important that this model becomes part of ongoing discussions about the geology of the Barents Sea and Svalbard. These thrust systems cannot be ignored anymore and should be top-priority targets in the next few years, e.g., to constrain plate tectonics models in the late Neoproterozoic to Cenozoic, or to explore for hydrocarbons or minerals, or for carbon storage, or even studying the hazard risk they present (e.g., Mitchell et al., 1990; Pirli et al., 2013). Regarding the 3D geometry of Timanian thrust systems, no 3D seismic data exist in the northern Barents Sea since it is not open for hydrocarbon exploration. However, high-resolution 3D seismic data on the Loppa High do further illustrate the model argued for in the present manuscript. These data being private and located in the southern Barents Sea, they will be described and discussed in a future manuscript.

Comment 9: agreed. This is a great point and the authors of the present manuscript agree that the present findings will lead to a new plate tectonics model for the Norwegian Arctic in the 650–0 Ma period. However, considering the recent discovery of the Timanian thrusts systems described in the present manuscript, it is without saying that a new plate tectonics model is beyond the scope of the paper. However, a new model is currently being worked out using GPlates and will follow up on the present manuscript's findings and its implications for plate tectonics reconstructions. Nevertheless, the authors of the present manuscript agree that a local and simple plate tectonics model should be included to the manuscript as suggested by Prof. Doré. Following Prof. Doré's recommendation, the authors of the present manuscript propose to include such a model as Figure 8.

Comment 10: agreed.

Comment 11: disagreed. This would overcrowd a figure already crowded with information. In addition, the age of Timanian fingerprints in other Arctic areas is not the point of the manuscript.

Comment 12: agreed.

Comment 13: agreed.

Comment 14: labelling each seismic section in Figure 1 would overcrowd the figure with trivial information. Instead, it is possible to obtain the name of each seismic section from the main author or from the Norwegian Petroleum Directorate. The authors of the present manuscript also feel that mentioning which specific studies did inspect which specific seismic lines in the Barents Sea would lead to a significant amount of irrelevant text. It is safe to assume that each previous study referenced in the present manuscript does include an overview of the database it used to support its own conclusions. If Dr. Rimando has any particular suggestion of previous works not acknowledged in the present study, the authors of the present manuscript welcome any addition, provided that it adds to the manuscript and allows further discussion. To the knowledge of the authors of the present manuscript, all seismic lines in the northern Norwegian Barents Sea are part of the DISKOS database. Again, if Dr. Rimando is aware of any data or contribution not acknowledged or discussed but should have, the authors of the present manuscript would welcome its addition to the manuscript.

The authors of the present manuscript feel that adding the "locations of previous studies which were discussed to provide evidence of the lateral extent of these Timanian thrusts" by, e.g., adding a frame for each study's extent in Figure 1, is irrelevant and would overcrowd an already crowded figure. If the reader is interested in the extent of a previous study or in the database used in a previous study, she/he should refer to the associated publication and/or contact the relevant author(s) if needed.

Comment 15: agreed. The Ellesmerian Orogeny, though believed not to have occurred by the lead author, is still commonly thought to be part of the geological history of Svalbard. In the present manuscript, it is only mentioned in the introduction and geological setting sections, and in the discussion where it is refuted as a possible cause of the accretion of Svalbard's basement terranes. However, it does not constitute the focus of the present manuscript and is therefore not discussed further. This issue will nevertheless be addressed in two manuscripts in preparation.

Comment 16: "top-SSW" is standard, as much as "top-to-the-SSW" is. The former requires fewer words and space and is not less explanatory. The authors of the present manuscript will of course update the manuscript if both reviewers and the editor judge it easier to read and comprehend for the reader.

Comment 17: agreed.

Comment 18: agreed. The strike and trend of geological structures will be added where appropriate but are already stated by dip direction, which are more informative because inform about the trend/strike and dip of the associated geological structure (e.g., "…-dipping"). However, the stress directions and changes of stress direction are mostly speculative and still a matter of debate in most cases. In addition, local variations exist (e.g., Brøggerhalvøya segment of the West Spitsbergen Fold-and-Thrust Belt which trends WNW–ESE, i.e., oblique to the rest of the fold-and-thrust belt). Stress directions will therefore not be added.

Comment 19: agreed. Yes indeed, it is a combination of "basement structure" and "deep-seated". If this is not correct, it may be rephrased of course.

Comment 20: agreed. The labels of seismic sections in Figure 1 are erroneous.

Comment 21: agreed. The label in Figure 2 was rephrased to "Max. horizontal stress".

Comment 22: agreed. However, these reflections need to be displayed in the same scheme (color/pattern) as their counter-parts in other units.

Comment 23: agreed.

Comment 24: agreed. The faults the present study deals with are actually oblique-slip. However, it is neither possible nor useful for the present manuscript to establish/discuss this issue. As mentioned by Dr. Rimando, it is important to establish the continuity of Timanian structures first. The oblique-slip character of the fault systems discussed in the present manuscript will be discussed in two other manuscripts. Two major lines of evidence suggest oblique-slip kinematics: (1) recent (2008–2019) deep (c. 15–16 kilometers) earthquakes in Storfjorden erroneously ascribed to a putative NE–SW-striking fault in Storfjorden suggest recent–ongoing sinistral-reverse oblique-slip movements along the KCFZ, BFZ, and KDFZ, and (2) major N–S-trending, Caledonian and Devonian basement ridges (e.g., Atomfjella Antiform; Witt-Nilsson et al., 1998) are offset left-laterally by c. 10–25 kilometers and in a reverse (top-SSW) fashion by c. 5–5.5 kilometers across the KCFZ.

Comment 25: agreed. However, this pre-print is already accessible at the following link: https://www.researchgate.net/publication/349124816_Devonian-Carboniferous_collapse_and_segmentation_of_the_Billefjorden_Trough_and_Eurekan_inversion -overprint_and_strain_partitioning_and_decoupling_along_inherited_WNW-_ESE-striking_faults?_sg=qtceO8VLbOUVZh5i5AT30gZbnAY8wO5q4mbX_u98eKImEuLQS8aOqk 0mc6guKuoXeagQlv1F3v9ZoAwHOfdtHSpw5RoUCAQcOUcC6Usc.EPUrQi5OpABDFpUhL XMyvgdMKcNL97-WXB5QVOsPliueVegj9fNgWuQ8QAiKNQrv-ojIEvVkbheFo1nTMDjVcg. The EGU guidelines state that "Works "submitted to", "in preparation", "in review", or only available as preprint should also be included in the reference list".

Comment 26: agreed. Figure 5b is a slope map of gravimetric data. This should be specified.

Comment 27: agreed.

Comment 28: agreed. The authors of the present manuscript are not allowed to transfer data from the DISKOS database directly to another party. Although the data are publicly accessible, the concerned party should submit inquiries to the Norwegian Petroleum Directorate.

Comment 29: agreed.

**3. Changes implemented**

Comment 1: none required by the reviewer.

Comment 2: modified Fig. 1b in the present manuscript to include the complete seismic database as in the attached Figure 1.

Comment 3: none.

Comment 4: "Speculative" portions of the faults (i.e., portions of the faults that are not demonstrated and argued for in the present study, but that are known from other ongoing manuscripts and studies) were added as dotted yellow lines in Figure 1b and c. Also see response to comment 2.

Comment 5: none.

Comment 6: awaiting further instructions from the editor and the two reviewers.

Comment 7: none.

Comment 8: none.

Comment 9: included a new Figure 8.

Comment 10: added "Hs" to Figure 1c and "Hs: Hornsund; " line 1535, "Norway" to Figure 1b, and "Pearya" in Figure 1a.

Comment 11: none.

Comment 12: added abbreviations of all major fault zones to the text lines 71, 86, 87, 88, 90, 135, 136, and 308–310.

Comment 13: added scale bars and north arrow (or "North Pole") labels to Figure 1a–c.

Comment 14: none but may include reference to other studies if Dr. Rimando has any specific study in mind that should be cited in the present manuscript.

Comment 15: none required by the reviewer.

Comment 16: may update the text if judged necessary by the reviewers and editor.

Comment 17: added "NE terrane", "SW terrane" and "NW terrane" labels in Figure 2.

Comment 18: added "NNE-dipping" lines 86, 89, 91, 119, 121, 123, and 152, "gently north-plunging" lines 88, and 142–143, "N–S-trending" lines 142, 144, and 145–146, and "N–S-striking" lines 135, and 193, and "E–W-trending" line 172.

Comment 19: may be adjusted into "deep-seated" or "basement structure" if incorrect use of English language. Awaiting further instructions from the editor and reviewers.

Comment 20: changed the labels of seismic sections in Figure 1.

Comment 21: rephrased label to "maximum horizontal stress".

Comment 22: none.

Comment 23: added a white dashed frame in Figure 1b showing the location of potential field data in Figure 5, and "(see location as a dashed white frame in Figure 1b)" line 1572.

Comment 24: none required by the reviewer.

Comment 25: none required by the reviewer.

Comment 26: added "for gravimetric data" lines 542–543 and "of gravimetric data" line 573.

Comment 27: changed from passive to active form lines 18, 21–22, 57, 92–97, 131, 139–142, 145, 216–219, 225–229, 232–233, 422–428, 530–534, 650–653, and 797–801.

Comment 28: removed "The complete seismic study is also available from the corresponding author upon request." lines 1113–1114.

Comment 29: none required by the reviewer.

**4. Additional changes implemented**

-Lines 80–81: added ", and imply that the Norwegian Barents Sea and Svalbard basement may contain Timanian structures overprinted during later (e.g., Caledonian) deformation events" for clarity.

-Lines 94–100: split the sentence into two and partly rewrote it to make it easier to read.

-Line 103: added "by future research" for clarity.

-Lines 1004–1005: split the sentence into two and added ". If correct, a Timanian origin for these structures would" to make it more readable.

-Lines 1055 and 1064: added reference to the new Figure 8 as a consequence to Prof. Doré comment 6.

[revised manuscript text omitted]

---

## Referee Report (RR1)

Comments on the "Impact of Timanian thrust systems on the late Neoproterozoic-Phanerozoic tectonic evolution of the Barents Sea and Svalbard" by Jean-Baptiste Koehl et al (se-2021-71)

This an interesting manuscript proposing an alternative model for the accretion of western Barents Sea and Svalbard. As a non-specialist in Timanian orogeny nor in Svalbard/Artic geology, I found it very challenging in comparing existing models, but well worth being published in Solid Earth. Hence, my comments are mainly regarding the methods used and presented observations/interpretations. This study is based on the geophysical methods, that are the interpretation of seismic reflection profiles, total gravity and magnetic field anomalies.

The study area records a very complex geology with several deformation phases (four compressional and two extensional) took place that are overprinting each other as it is stated by the authors. In addition, the study area is located in the offshore with only three wells presented. Therefore, this study greatly relies on the geophysical methods. In such a case and in order to increase the accuracy of the presented interpretation I would start by carefully characterizing (as much as available data and previous studies allows) geophysical signature (here magnetic, gravity and seismic reflection) of the known Timanian structures onshore/close to the shore.

In this setting, Trollfjorden-Komagelva Fault Zone (TKFZ) as a well-known Timanian structure onshore northern Norway could be the best candidate as the geophysical onshore-offshore data in northern Norway are fairly accessible. It would be very interesting and helpful if authors quantify geophysical character of the TKFZ, its spatial relationship to the overprinting Caledonian and younger events and describe how this structure extends to the offshore then use that as an analogue for the study area.

In potential filed data one would try to used different filtering techniques and attributes in order to separate deeper (presumably older) structures form the shallower (younger) structures and then study spatial development of the interested structures. Without an attempt to separate relative depth of causative bodies observed in potential filed data it is very hard to identify structures related to different tectonic events. I am not sure if we can simply interpret all E-W striking anomalies observed on total magnetic field and gravity data as Timanian and all N-S to NNE-SSW as Caledonian. I do agree and acknowledge that the main trends can be identified in gravity and magnetic data, but author should also consider and discuss alternative interpretations for observed trends, especially considering spatial geometry of structures over several hundreds of kilometers. In the next step, observations from the potential field data can be compared with the seismic reflection data.

As it is mentioned by the authors, e.g. Barrère et al. 2009, Gernigon et al. 2014 & 2018, and the ATLAS, Geological History of the Barents Sea (Geological survey of Norway, 2009, not cited by the authors) have done such a methodology in different parts of the Barents Sea concluding that post-Timanian events (mainly Caledonian) overprinting the Timanian structures and continuation of Timanian structures is only identified in northern Norway (TKFZ). I understand that above mentioned studies might have not been aiming for mapping the westward extension of the Timanian structures, but I think it would be of interest if authors consider discussing similarities and differences between potential filed data interpretation in this study and previous ones.

Another major Timanian structure identified in Novaya Zemlya Island is the Baidaratsky Fault Zone (BaFZ, shown in Figs. 1 and 5). BaFZ is mapped onshore Novaya Zemlya as awide (ca. 30 km) fault zone (e.g. Lopatin et al., 2001; Korago et al., 2004). Korago 2004 in their Fig. 8 show a NW continuation of BaFZ (dashed line) and state that "presumably" BaFZ continues NW into the eastern Barents Sea.

However, Korago et al., 2004 did not carried any offshore studies in this regard and refer to Lopatin 2001. While Lopatin et al. 2001 also did not studied western offshore Novaya Zemlya and only show the location of BaFZ onshore. Therefore, based on Lopatin et al., 2001 and Korago et al., 2004 it is really difficult to conclude any NW extension of BaFZ into the eastern Barents Sea. I understand that accessing geophysical data in eastern Barents Sea is challenging, however, some across border studies (e.g. ATLAS, Geological History of the Barents Sea, 2009, Geological Survey of Norway) are available and could be used in gravity and magnetic analysis and interpretations. Looking at filtered magnetic and gravity and presented derivatives presented in their Chapter 2 (IMAGING DEEP STRUCTURES BENEATH THE SURFACE) I can recognize E-W to ENE-WSW oriented structures onshore and offshore south of Novaya Zemlya Island extending SE into the Russian main land (Pechora Basin?). Farther west from Novaya Zemlya and into the central Barents Sea main structures are N-S striking. Based on above, I have difficulties tracing BaFZ all the way into the western Barents Sea and link it to the E-W structures south of Olga Basin shown in Fig.1b. I do agree that in the western Barents Sea there are structures orienting E-W and ENE-WSW, but also there are N-S and NE-SW structures. It would be very helpful if authors could explain such a complexity in the western Barents Sea and westward extension of BaFZ specially across the areas with very strong N-S orienting magnetic and gravity signature.

I would assume that westward extension of identified thrust zones into the onshore Svalbard is based on the gravity and magnetic data. Looking at Fig.5, onshore Svalbard is at the edge of the dataset and it is not really possible to see any trends, while filtered magnetic and gravity maps shown in the ATLAS, Geological History of the Barents Sea, 2009 covers the entire Svalbard and its western offshore, showing N-S trends being very pronounced. I would suggest authors compare their observations with above mentioned reference and discuss potential differences and similarities observed. Also, as Fig. 1b shows there are seismic profiles available on the western offshore Svalbard, do those seismic profiles have also been studied? Do they show extension of identified thrust zones across Svalbard? Sine authors argue that Timanian structures onshore Svalbard are unnoticed because of the remoteness of the area and the strongly eroded character of the area, showing the extension of Timanian structures west of Svalbard could provide an additional proof for the presence of Timanian structures across the Svalbard.

I assume that the shown seismic reflection profiles are the best examples from many other studied and interpreted profiles. However, the quality of presented profiles really does not allow readers to attempt interpreting profiles, even higher quality version of seismic profiles made available by authors did not help. I would suggest authors to use higher quality and less noisy profiles (if available), in the shown profiles I can see some intra-basement trends, but I also can add in much more patterns. As an example, along the profile shown in Fig.3b lost of patterns are not interpreted in the center of the profile, what would those reflections represent? In addition, confirming thrust zones dip direction (since dip directions mentioned in the text are apparent dip) it would be much more convincing if author show at least one profile parallel to 3a and 3c farther east as fig.1b shows that are more profiles available east of 3a and 3c.

As profile 3b is semi-perpendicular to main Caledonian N-S trend, it would be very interesting if authors consider interpreting Caledonian structures along profile 3b and show/discuss the spatial relationship between Timanian and Caledonian structures.

Authors claim that that thicker Precambrian basement rocks shows higher Bouguer anomaly values (lines 532-535) and take this as an evidence for the thrusting causing thickening of basement rock into the footwall of thrust faults. Looking at profile 3a, the southern parts of the profile shows thickest Devonian-Permian sedimentary rocks and thinner Precambrian basement rocks. Such a configuration should be reflected as low Bouguer anomaly (thick sedimentary rocks) while shown gravity anomaly

profile in the lower panel show high gravity values. On the opposite end of the same profile (Fig. 3a) where the Precambrian units are thicker gravity anomaly profile shows very low values. Same inconsistency also appears along profiles 3b (in the center) and 3c (to the north). This is confusing, please consider clarifying.

Closest well utilized for well-seismic tie in the study is the well Hopen-2 which is located 40-45 km north of profile shown in Fig. 3b. According to Harald and Kelly 1997 and Anell et al. 2014 well Hopen-2 is drilled into Late Carboniferous sedimentary rocks and the top basement is not reached. Please consider briefly explaining how boundaries between Precambrian, Cambrian-Silurian, Devonian-Mississippian and Devonian-Permian are identified and interpreted.

In the proposed model shown in Fig. 7, I am wondering when a several km thick shear/thrust zone inherited form the Timanian event exist (Fig. 7a) why such a structure is not simply reactivated as strike-slip fault/shear zone and instead it is folded and cross-cut by Caledonian structures? Could authors back up this model with natural cases or modeling studies? A discussion elaborating this would be of interest.

In general, this is a well-written article presenting geophysical evidence for and further highlighting existing models proposing westward extension of Timanian structures across the Barents Sea. The study also discusses pre-Caledonian plate tectonics implications of such a configuration that it might be of great interest for Solid Earth readers. I believe the paper is very interesting and can be published after addressing my comments. I would be happy to further discuss my comments and look forward to seeing this manuscript being published.

Best regards,

Hamed Fazlikhani

---

## Author Response (AR2)

**Reply to Hamed Fazlikhani**

Dear Dr. Fazlikhani,

thank you very much for your input on the manuscript, it is highly appreciated. Here is our reply to your comments. We hope the changes we implemented improve the shortcomings of the manuscript highlighted by your comments and suggestions. Please do not hesitate to contact us shall this not be the case for some comments.

**1. Comments from Dr. Fazlikhani**

Comment 1: This an interesting manuscript proposing an alternative model for the accretion of western Barents Sea and Svalbard. As a non-specialist in Timanian orogeny nor in Svalbard/Artic geology, I found it very challenging in comparing existing models, but well worth being published in Solid Earth. Hence, my comments are mainly regarding the methods used and presented observations/interpretations. This study is based on the geophysical methods, that are the interpretation of seismic reflection profiles, total gravity and magnetic field anomalies.

Comment 2: The study area records a very complex geology with several deformation phases (four compressional and two extensional) took place that are overprinting each other as it is stated by the authors.

Comment 3: In addition, the study area is located in the offshore with only three wells presented. Therefore, this study greatly relies on the geophysical methods. In such a case and in order to increase the accuracy of the presented interpretation I would start by carefully characterizing (as much as available data and previous studies allows) geophysical signature (here magnetic, gravity and seismic reflection) of the known Timanian structures onshore/close to the shore.

Comment 4: In this setting, Trollfjorden-Komagelva Fault Zone (TKFZ) as a well-known Timanian structure onshore northern Norway could be the best candidate as the geophysical onshore-offshore data in northern Norway are fairly accessible. It would be very interesting and helpful if authors quantify geophysical character of the TKFZ, its spatial relationship to the overprinting Caledonian and younger events and describe how this structure extends to the offshore then use that as an analogue for the study area.

Comment 5: In potential filed data one would try to used different filtering techniques and attributes in order to separate deeper (presumably older) structures form the shallower (younger) structures and then study spatial development of the interested structures. Without an attempt to separate relative depth of causative bodies observed in potential filed data it is very hard to identify structures related to different tectonic events. I am not sure if we can simply interpret all E-W striking anomalies observed on total magnetic field and gravity data as Timanian and all N-S to NNE-SSW as Caledonian.

Comment 6: I do agree and acknowledge that the main trends can be identified in gravity and magnetic data, but author should also consider and discuss alternative interpretations for observed trends, especially considering spatial geometry of structures over several hundreds of kilometers. In the next step, observations from the potential field data can be compared with the seismic reflection data.

Comment 7: As it is mentioned by the authors, e.g. Barrère et al. 2009, Gernigon et al. 2014 & 2018, and the ATLAS, Geological History of the Barents Sea (Geological survey of Norway, 2009, not cited by the authors) have done such a methodology in different parts of the Barents Sea concluding that post-Timanian events (mainly Caledonian) overprinting the Timanian structures and continuation of Timanian structures is only identified in northern Norway (TKFZ).

Comment 8: I understand that above mentioned studies might have not been aiming for mapping the westward extension of the Timanian structures, but I think it would be of interest if authors consider discussing similarities and differences between potential filed data interpretation in this study and previous ones.

Comment 9: Another major Timanian structure identified in Novaya Zemlya Island is the Baidaratsky Fault Zone (BaFZ, shown in Figs. 1 and 5). BaFZ is mapped onshore Novaya Zemlya as awide (ca. 30 km) fault zone (e.g. Lopatin et al., 2001; Korago et al., 2004). Korago 2004 in their Fig. 8 show a NW continuation of BaFZ (dashed line) and state that "presumably" BaFZ continues NW into the eastern Barents Sea. While Lopatin et al. 2001 also did not studied western offshore Novaya Zemlya and only show the location of BaFZ onshore. Therefore, based on Lopatin et al., 2001 and Korago et al., 2004 it is really difficult to conclude any NW extension of BaFZ into the eastern Barents Sea.

Comment 10: I understand that accessing geophysical data in eastern Barents Sea is challenging, however, some across border studies (e.g. ATLAS, Geological History of the Barents Sea, 2009,

Geological Survey of Norway) are available and could be used in gravity and magnetic analysis and interpretations.

Comment 11: Looking at filtered magnetic and gravity and presented derivatives presented in their Chapter 2 (IMAGING DEEP STRUCTURES BENEATH THE SURFACE) I can recognize E-W to ENE-WSW oriented structures onshore and offshore south of Novaya Zemlya Island extending SE into the Russian main land (Pechora Basin?).

Comment 12: Farther west from Novaya Zemlya and into the central Barents Sea main structures are N-S striking. Based on above, I have difficulties tracing BaFZ all the way into the western Barents Sea and link it to the E-W structures south of Olga Basin shown in Fig.1b. I do agree that in the western Barents Sea there are structures orienting E-W and ENE-WSW, but also there are N-S and NE-SW structures. It would be very helpful if authors could explain such a complexity in the western Barents Sea and westward extension of BaFZ specially across the areas with very strong N-S orienting magnetic and gravity signature.

Comment 13: I would assume that westward extension of identified thrust zones into the onshore Svalbard is based on the gravity and magnetic data.

Comment 14: Looking at Fig.5, onshore Svalbard is at the edge of the dataset and it is not really possible to see any trends, while filtered magnetic and gravity maps shown in the ATLAS, Geological History of the Barents Sea, 2009 covers the entire Svalbard and its western offshore, showing N-S trends being very pronounced. I would suggest authors compare their observations with above mentioned reference and discuss potential differences and similarities observed.

Comment 15: Also, as Fig. 1b shows there are seismic profiles available on the western offshore Svalbard, do those seismic profiles have also been studied? Do they show extension of identified thrust zones across Svalbard? Sine authors argue that Timanian structures onshore Svalbard are unnoticed because of the remoteness of the area and the strongly eroded character of the area, showing the extension of Timanian structures west of Svalbard could provide an additional proof for the presence of Timanian structures across the Svalbard.

Comment 16: I assume that the shown seismic reflection profiles are the best examples from many other studied and interpreted profiles. However, the quality of presented profiles really does not allow readers to attempt interpreting profiles, even higher quality version of seismic profiles made available by authors did not help.

Comment 17: I would suggest authors to use higher quality and less noisy profiles (if available), in the shown profiles I can see some intra-basement trends, but I also can add in much more patterns. As an example, along the profile shown in Fig.3b lost of patterns are not interpreted in the center of the profile, what would those reflections represent?

Comment 18: In addition, confirming thrust zones dip direction (since dip directions mentioned in the text are apparent dip) it would be much more convincing if author show at least one profile parallel to 3a and 3c farther east as fig.1b shows that are more profiles available east of 3a and 3c.

Comment 19: As profile 3b is semi-perpendicular to main Caledonian N-S trend, it would be very interesting if authors consider interpreting Caledonian structures along profile 3b and show/discuss the spatial relationship between Timanian and Caledonian structures.

Comment 20: Authors claim that that thicker Precambrian basement rocks shows higher Bouguer anomaly values (lines 532-535) and take this as an evidence for the thrusting causing thickening of basement rock into the footwall of thrust faults. Looking at profile 3a, the southern parts of the profile shows thickest Devonian-Permian sedimentary rocks and thinner Precambrian basement rocks. Such a configuration should be reflected as low Bouguer anomaly (thick sedimentary rocks) while shown gravity anomaly profile in the lower panel show high gravity values. On the opposite end of the same profile (Fig. 3a) where the Precambrian units are thicker gravity anomaly profile shows very low values. Same inconsistency also appears along profiles 3b (in the center) and 3c (to the north). This is confusing, please consider clarifying.

Comment 21: Closest well utilized for well-seismic tie in the study is the well Hopen-2 which is located 40-45 km north of profile shown in Fig. 3b. According to Harald and Kelly 1997 and Anell et al. 2014 well Hopen-2 is drilled into Late Carboniferous sedimentary rocks and the top basement is not reached.

Comment 22: Please consider briefly explaining how boundaries between Precambrian, Cambrian-Silurian, Devonian-Mississippian and Devonian-Permian are identified and interpreted.

Comment 23: In the proposed model shown in Fig. 7, I am wondering when a several km thick shear/thrust zone inherited form the Timanian event exist (Fig. 7a) why such a structure is not simply reactivated as strike-slip fault/shear zone and instead it is folded and cross-cut by Caledonian structures? Could authors back up this model with natural cases or modeling studies? A discussion elaborating this would be of interest.

Comment 24: In general, this is a well-written article presenting geophysical evidence for and further highlighting existing models proposing westward extension of Timanian structures across the Barents Sea. The study also discusses pre-Caledonian plate tectonics implications of such a configuration that it might be of great interest for Solid Earth readers. I believe the paper is very interesting and can be published after addressing my comments. I would be happy to further discuss my comments and look forward to seeing this manuscript being published.

**2. Author's reply**

Comment 1: agreed.

Comment 2: agreed, though it is now becoming clear that one of the contractional events, the Ellesmerian Orogeny, never occurred in Svalbard and the Barents Sea (e.g., Koehl, 2021).

Comment 3: agreed. However, the present manuscript is the first account of the seismic character of Timanian faults next to the shore of Svalbard. Timanian magnetic and gravimetric anomalies in the northern Norwegian Barents Sea were first described in Klitzke et al. (2019). In northern Norway/northwestern Russia, the Trollfjorden–Komagelva Fault Zone/Central Timan Fault (i.e., the Timanian front thrust) and related anticlines on the Varanger Peninsula (e.g., dome-shaped Ragnarok Anticline; Siedlecka and Siedlecki, 1971) and in Russia (e.g., WNW–ESE-trending Mikulkin Antiform on the Kanin Peninsula; Lorenz et al., 2004; see also their figures 5 and 6) are, as shown in figure 5 in the present manuscript, characterized by positive WNW–ESE-trending magnetic and gravimetric anomalies that can be traced from Varanger Peninsula in northeasternmost Norway to the Kanin Peninsula in northwestern Russia. The magnetic anomaly related to the Trollfjorden–Komagelva Fault Zone in northeastern Norway is also shown in Nasuti et al. (2015) and Koehl et al. (2019). In addition, ongoing work suggest that the Sørøya–Ingøya shear zone, a presumed Caledonian thrust first described in Koehl et al. (2018), actually represents the folded continuation of the Trollfjorden–Komagelva Fault Zone, which was folded and partly reactivated as a thrust during the Caledonian Orogeny (Koehl, in prep.). Thus, one may view the geometry of the Sørøya–Ingøya shear zone on seismic data in Koehl et al. (2018) as an analog to Timanian thrust systems in the northern Barents Sea. In northwestern Russia, the seismic character of major Timanian thrusts is shown in various studies, including notably Kostyuchenko et al. (2006, their figure 17 notably). However, studies onshore northwestern Russia and northeasternmost Norway are still far away from the northern Norwegian Barents Sea. Thus, the authors of the

present manuscript feel that it is more appropriate to describe the structures they identified first, and to compare them with known examples of Timanian faults in adjacent areas in the discussion. Noteworthy, the correlation of Kostyuchenko et al. (2006) of magnetic data and Timanian structures is unambiguous: "The drillholes into the basement beneath the Pechora Basin […] demonstrated that the very strong magnetic anomalies of the Pechora Zone outlined by the 'Pre-Pechora' Faults (shown in Figs 11 and 18) coincided with a belt of volcanic and volcano-sedimentary rocks with major gabbro-diorite intrusions and granites", i.e., that WNW–ESE-trending magnetic anomalies in northwestern Russia can be directly correlated to volcanic belts bounded by Timanian faults (see also their figures 2, 3 and 18). Their correlation of Timanian structures with gravimetric anomalies is also unambiguous: "The grade of metamorphism correlates well with the gravity data. Thus, strong positive gravity anomalies occur over the Kanin Peninsula [where the Mikulkin Antiform of Lorenz et al. (2004) occurs], whereas much less positive anomalies cover the general area of the Timan Range", and they easily correlated thickened dense basement with high metamorphic grade to positive gravimetric anomalies. The authors of the present manuscript concede that these correlation onshore northwestern Russia could be further specified in the manuscript.

Comment 4: see response to comment 3. The western continuation of the Trollfjorden-Komagelva Fault Zone has been extensively debated in the past few years. Initially the fault was thought to proceed in a rectilinear fashion offshore (Gabrielsen and Færseth, 1989; Gabrielsen et al., 1990; Roberts et al., 2011). However, recent studies of this fault complex on 2D and 3D seismic data (Koehl et al., 2018), magnetic data and fieldwork (Koehl et al., 2019) suggest that it is not the case. Notably, there is no fault on 3D seismic data in the footwall of the Måsøy Fault Complex where the Trollfjorden–Komagelva Fault Zone is believe to proceed offshore (Koehl et al., 2018 their figure 8). This fault is now believe to be folded and to continue as a NE–SW-trending thrust system (Koehl, in pre.; see Sørøya–Ingøya shear zone in Koehl et al., 2018). The magnetic signature of the fault is described in Nasuti et al. (2015) and Koehl et al. (2019) and correlates with positive magnetic anomalies related to Mississippian dolerite dykes intruded along WNN–ESE-striking segments of the fault complex (Roberts et al., 1991; Lippard and Prestvik, 1997). The gravimetric character of the fault is still unclear (essentially not discussed in existing literature), but based on the new correlation of the Trollfjorden–Komagelva Fault Zone with its folded continuation offshore to the west (Sørøya–Ingøya shear zone of Koehl et al., 2018), the fault correlated with a

positive gravimetric anomaly that bends in the same way as the fault in the west offshore (Skilbrei et al., 2000). However, since this work is still being written into a manuscript, it does not sound natural to include it in the present manuscript. The spatial interaction of the Trollfjorden–Komagelva Fault Zone with Caledonian structures is illustrated by the dome-shaped geometry of the Ragnarokk Anticline of Siedlecka and Siedlecki (1971) on the Varanger Peninsula (refolding of a Timanian, thrust-related anticline during the Caledonian Orogeny; see present manuscript lines 807–810 and 843–847).

Comment 5: disagreed. Timanian faults formed in the latest Neoproterozoic and are several kilometers (to several tens of kilometers thick; see seismic sections in figure 3). Later on, these faults controlled the formation of new faults and folds during the entire Phanerozoic. The same (Timanian and Caledonian) trends are therefore to be found at depths shallower than Top-basement (post-orogenic and future rift basins controlled by existing basement grains). Separating all depths in the magnetic and gravimetric datasets would imply assuming that each tectonic event affected only one layer of the crust and none of the underlying nor overlying layers. It is by disentangling the whole dataset (all levels of the crust influenced by Timanian structures) that one may resolve the issues approached by the present manuscript. It is not the aim of the authors of the present manuscript to interpret all (overall) WNW–ESE-trending magnetic and gravimetric anomalies as Timanian and all N–S- to NE–SW-trending anomalies as Caledonian, but as anomalies composed of Timanian structures and all younger superimposed structures that localized along these existing Timanian structures (Caledonian reactivation, late Paleozoic extensional basins, possibly Mesozoic basins, early Cenozoic basins and inversion, and possibly in the west late Cenozoic rift basins) and that, therefore, formed with the same trend. The sum of all these superimposed structures developed along the dominant two structural trends (Timanian and Caledonian) is believed to have further anchored the two structural trends in the crust, which therefore shows very nicely on potential field data at present.

Comment 6: the only structures with WNW–ESE strikes in northeasternmost Norway and northwestern Russia are all related to the Timanian Orogeny and to reactivation/overprinting of Timanian structures.

Comment 7: agreed. It is appropriate to add the Geological Atlas of the Barents Sea to the present manuscript's reference list. Importantly, the NE–SW-trending seismic profile in the Russian Barents Sea (profile C–D, pp. 53 in Smelror et al., 2009; location of the profile shown pp. 43)

clearly shows the Baidaratsky Fault Zone in the central part with a similar configuration as in figure 3d in the present manuscript, i.e., a major, low-angle basement-seated fault inverted as a listric normal fault that localized the deposition of a Paleozoic basin. However, it is incorrect that the Timanian trend was identified exclusively in northern Norway. Recent work off the coasts of Finnmark now clearly show that Timanian grain is present in the crust of the southeastern Norwegian Barents Sea too and had a tremendous impact on subsequent tectonic events by controlling the formation of subsequent fault and basins (Hassaan et al., 2020a, 2020b, 2021; Hassaan, 2021).

Comment 8: agreed. This is done lines 766–768, 817–821, and 1072–1078 for the Marello et al (2010) and Barrère et al. (2011) studies, and lines 606–610, 720–730, and 784–790 for the Gernigon and Brönner (2012) and Gernigon et al. (2014) studies. Notably, lines 817–821: "Furthermore, Barrère et al. (2011) suggested that basins and faults in the southern Norwegian Barents Sea are controlled by the interaction of Caledonian and Timanian structural grain, and Marello et al. (2010) argued that elbow-shaped magnetic anomalies reflect the interaction of Caledonian and Timanian structural grains in the Barents Sea, potentially as far west as the Loppa High and the Bjørnøya Basin", the authors of the present manuscript discuss the geometry of magnetic and gravimetric anomalies in the perspective of the reworking of Timanian grain during the Caledonian Orogeny, which was also previously inferred by previous studies in the southern and central Norwegian Barents Sea, i.e., similar findings. The main difference with previous studies is that the present study goes further because the present manuscript includes interpreted seismic sections in the northern Barents Sea and onshore–nearshore Svalbard with well tie showing clear thrust fault geometries (Figure 3).

Comment 9: disagreed. Lopatin et al. (2001) present their interpretation of a nearby offshore seismic profile in their figure 1 (figure caption: "Geological section after offshore seismic profiling"). They also mention in their abstract that their data include "seismic profiling". Thus, they did investigate the western continuation of the Baidaratsky fault zone west of Novaya Zemlya with data available to them. The Lopatin et al. (2001) is then cited by Korago et al. (2004) to be the study that has produced the work on seismic data to map the Baidaratsky Fault Zone in the Russian Barents Sea, although the short Lopatin et al. (2001) article only shows the extent of the fault onshore and nearshore: "The Baidaratsky fault zone is expressed by a series of

strike-slip faults, which can be seen on the seismic records in the Barents Sea (Lopatin *et al.* 2001)" (second paragraph after the abstract in Korago et al., 2004).

Comment 10: agreed. It is not possible to access data covering Russian territory outside Russia. We also agree that the Smelror et al. (2009; Geological Atlas of the Barents Sea) should be cited in the present manuscript in referred to in the text when discussing our interpretation. See also response to comment 7.

Comment 11: agreed.

Comment 12: agreed. The "complexity" mentioned by Dr. Fazlikhani is part of the issue raised and discussed by the present manuscript. The magnetic signature of the Baidaratsky Fault Zone locally disappears in the central Russian Barents Sea because this portion of the Barents Sea was mildly deformed into large synclines during Caledonian contraction because located away from the collision front, i.e., magnetic signal of Timanian faults pushed down and more difficult to trace at the location of major Caledonian synclines (this will be added to the discussion). In the west, i.e., closer to the Caledonian collision front, Timanian faults were extensively reworked, but not to the point of not being able to identify them as seen on seismic data (Figure 3 in the present manuscript). The present manuscript further highlights that Timanian faults are being reactivated/overprinted gradually less and less in the plate interior as shown by the ongoing reactivation of Timanian grain in the Fram Strait and Storfjorden (offset of seafloor in present manuscript Figure 3, and Koehl et al., 2021), whereas Timanian faults below the Olga Basin and in the central Barents Sea were last active in the late Paleozoic (Figure 3d and Smelror et al., 2009 their profile C–D pp. 53).

Comment 13: the prolongation of the WNW–ESE-striking thrust systems into eastern and central Spitsbergen is also based on seismic interpretation (see Koehl, 2021 and supplements S2c and S2d of the present manuscript).

Comment 14: the authors of the present manuscript have already interpreted magnetic and gravimetric data over the whole Svalbard Archipelago (see EGU Keynote by Koehl, 2020), which show clear evidences of continuation of Timanian grain across Svalbard (see for example slide 129 in Koehl, 2020). However, as mentioned in the present study and in our response to various comments, the structural setting along the western Barents Sea and western Spitsbergen margin is slightly more complicated because they are located adjacent to paleo-plate boundaries during the Caledonian Orogeny and Eurekan tectonic event, both of which reworked Timanian structures more than their counterparts farther east (e.g., from central–eastern Spitsbergen where Timanian

faults become relatively easy to trace and correlate). Thus, we consider that it is necessary to discuss the interpretation of magnetic and gravimetric data over Spitsbergen in a separate manuscript. This manuscript will also include bathymetric data around the Svalbard Archipelago and data from previous field campaigns, which do not fit in the present study and therefore warrant a separate manuscript. The following paragraph is a glimpse at the content of the hereby referred manuscript that is currently under writing.

Gravimetric data over Svalbard (see Figure 1 attached below) show a major change in gravimetric signal between northern and southern Svalbard exactly at the location of the mapped continuation of the Kongsfjorden–Cowanodden fault zone (high gravimetric anomalies in the north and low in the south). Notably, the low gravimetric anomaly correlated to the Central Tertiary Basin (Eurekan foreland basin) appears to continue across the Kongsfjorden–Cowanodden fault zone but with a significantly reduced width (dotted white lines). Since there are no Cenozoic sedimentary rocks equivalent to those of the Central Tertiary Basin in northwestern Spitsbergen (north of Kongsfjorden), we conclude that the anomaly is partly reflecting basement grain and that this grain (most likely a major N–S- to NNW–SSE-trending syncline) matches the geometry of the Central Tertiary Basin. The abrupt decrease in width of the major syncline suggests that it is offset in a top-SSW reverse manner and, thus, that the Kongsfjorden–Cowanodden fault zone continues all the way to western Spitsbergen. Moreover, tilt-derivative of magnetic data over Spitsbergen clearly show that N–S-trending anomalies are laterally offset by E–W- to NW–SE-trending lineaments (Koehl, 2020 pp. 129).

Comment 15: authors of the present manuscript have interpreted the whole seismic database around Svalbard and found evidences supporting the continuation of WNW–ESE-striking Timanian thrust across the whole archipelago and even some continental fragments with Timanian shear zones in the Fram Strait (e.g., Koehl, 2020 pp. 162–165). Also see response to comment 14 and Figure 1 attached below. Again, these will be published in a separate manuscript in order to adequately address their implications for the opening of the Fram Strait and ongoing processes such as earthquake cycles and methane seepage.

Comment 16: the authors of the present manuscript have provided high-resolution versions of the figures at DataverseNO: [dataverse.no/dataset.xhtml?persistentId=doi:10.18710/CE8RQH](dataverse.no/dataset.xhtml?persistentId=doi:10.18710/CE8RQH)). Notably, figure 3a, b and c are several hundreds of megabytes each and one may easily zoom in individual structures.

Comment 17: agreed. The authors of the present manuscript did not interpret every single structure on the seismic sections because there is simply not enough time to interpret them all. The interpreted structures displayed in Figure 3 in the present manuscript took overall three years to interpret. In addition, two years were necessary to interpret the whole dataset prior to making detailed interpretations as those shown in Figure 3 in the present manuscript. It is of course always possible to add to one's interpretation, but the authors of the present manuscript are confident that the presented structures are sufficient to support the argumentation and the conclusions detailed in the present manuscript. A lot of the reflections in the center of profile 3b represent N–S-trending, hundreds of meters wide Caledonian folds. It was however not possible to interpret them all due to time constraints. Interpreting them all would also be irrelevant if their interpretation does not add to the manuscript.

Comment 18: The present manuscript already includes such a N–S-trending seismic profile east of profiles 3a and 3c. The profile is shown in Figure 3d. We also note that all data are from the DISKOS database and are publicly accessible via contacting the Norwegian Petroleum Directorate.

Comment 19: agreed. These structures are described lines 365–382 and 414–442 and discussed lines 783–821, and 825–895 in the present manuscript. Caledonian structures are indeed interpreted in figure 3b and corresponds to the N–S-trending folds in lower Paleozoic basins and underlying basement.

Comment 20: agreed. It is the thickening of the denser portion of basement rocks (i.e., those with higher metamorphic grade, e.g., mylonites) that is thought to be responsible for the high gravimetric anomalies (see also clear correlation of high-grade Timanian metamorphic rocks with WNW–ESE-trending Bouguer anomalies in Kostyuchenko et al., 2006). This was clarified in the present manuscript following the response to comment 3. It is true that, in that specific instance (southernmost portion of profile 3a), the gravimetric anomaly further increases south of the thrust. The authors of the present manuscript do not argue that Timanian faults are the only features that may contribute to positive gravimetric anomalies in the Barents Sea. However, one may observe that the general correlation established between high-grade metamorphic rocks within Timanian thrust and positive gravimetric anomalies by the present manuscript is generally respected throughout the Barents Sea (Figure 3 in the present manuscript, Lorenz et al., 2004 and Kostyuchenko et al., 2006). Notably, the Kinnhøgda–Daudbjørnpynten fault zone correlates with a positive gravimetric anomaly (Figure 3a) even though another positive anomaly is found south

of the fault. Timanian faults are therefore major contributors to elevated Bouguer anomalies in the Barents Sea, but other features may, in places, also influence gravimetric anomalies (e.g., Caledonian folds and thrusts; Figure 5a).

Comment 21: agreed.

Comment 22: agreed. The boundaries between Devonian–Permian and Mesozoic sedimentary successions were tied to the three exploration wells mentioned in the present study for offshore parts of the study area. The boundary between Devonian–Mississippian and Pennsylvanian–Permian units onshore Svalbard are interpreted as a major unconformity truncating Devonian–Mississippian dykes (see Figure 3e). The boundary between Precambrian, lower Paleozoic and upper Paleozoic offshore are major unconformities that truncate underlying reflections and fold structures (e.g., Figure 3a, b and c).

Comment 23: agreed. The Timanian thrusts presented were oriented sub-orthogonal (c. 70 degrees) to the E–W principal stress during the Caledonian Orogeny in Svalbard. Thus one would expect that they were reactivated as strike-slip faults, which they partly did in repeated occasions, such as during the Caledonian Orogeny (e.g., Majka et al., 2008; Mazur et al., 2009; Faehnrich et al., 2020) and post-Caledonian Devonian collapse (e.g., Ziemniak et al., 2020). However, a reactivation simply and solely as strike-slip faults is not likely as the Timanian thrusts are low-angle faults and are therefore more prone to accommodating vertical movements. Hence, Caledonian E–W contraction produced more easily N–S-trending folds (vertical uplift and folding of rocks not hampered by any rocks upwards), which extended almost all the way to Novaya Zemlya (Figure 5), whereas partial strike-slip reactivation was restricted to areas proximal to the Caledonian collision front (e.g., western Spitsbergen; Majka et al., 2008; Mazur et al., 2009; Faehnrich et al., 2020; Ziemniak et al., 2020) because lateral transport of rocks from the Caledonian collision front towards the inner portions of the Barents Sea in the east was hampered by rock units constituting the crust of the Barents Sea, northern Norway, northwestern Russia and other adjacent areas. Therefore, despite a partial reactivation as strike-slip faults, these faults were also folded and locally overprinted by N–S-striking thrusts (e.g., in Nordmannsfonna; Figure 3e–f). This is what is illustrated in Figure 7 and it should be better explained in the discussion chapter in the present manuscript.

Comment 24: agreed.

**3. Changes implemented**

Comment 1: none recommended by the referee's comment.

Comment 2: none recommended by the referee's comment.

Comment 3: specified in the "Methods and datasets" section "bounding magmatic complexes and/or intruded by magmatic bodies" lines 243–244, and added reference to the work by Kostyuchenko et al. (2006) line 245. Added to the Introduction chapter reference to the Mikulkin Antiform on the Kanin Peninsula of Lorenz et al. (2004) lines 90–91 ("– and related Mikulkin Antiform"), line 125 ("(and associated thrust anticline, the Mikulkin Antiform)"), lines 241–242 ("(e.g., Mikulkin Antiform; Lorenz et al., 2004)"), lines 548–549 ("and associated Mikulkin Antiform"), and reference to Lorenz et al. (2004) lines 91, 126, 242, 560, 567, 727, 732, 735, 738–739, 748–749. Added "and its eastwards continuation, the Central Timan Fault (Lorenz et al., 2004; Kostyuchenko et al., 2006)" lines 734–735. Rewrote the sentence lines 736–740 into "In addition, the size of Timanian thrust systems and related thrust anticlines in the Timan Range and Kanin Peninsula (e.g., Central Timan Fault and Mikulkin Antiform) are comparable (≥ 3–4 seconds TWT thick thrusts and 5–15 kilometers wide thrust-related major anticlines; Lorenz et al., 2004 their figures 3 and 5; Kostyuchenko et al., 2006 their figure 17) to that of thrust and fold systems in the northern Norwegian Barents Sea and Svalbard (**Error! Reference source not found.**a and c–d).". Added "and associated major anticlines" line 745, ", 5–15 kilometers wide anticlines" lines 746–747, and "and fold system" line 751. Added "with high metamorphic grade" line 241 and reference to Kostyuchenko et al. (2006) line 242. Added ", possibly with higher metamorphic grade" lines 536–537 and "(i.e., higher metamorphic grade)" line 595. Also added Lorenz et al. (2004) to the reference list.

Comment 4: see response to comment 3.

Comment 5: none.

Comment 6: none recommended by the referee's comment.

Comment 7: reference to the Geological Atlas of the Barents Sea was added to the reference list. Added " This is supported by a similar configuration of the Baidaratsky Fault Zone and the Kongsfjorden–Cowanodden fault zone, including a basement-seated, low-angle thrust geometry of both faults and inversion as a normal fault and deposition of several seconds (TWT) thick sedimentary strata in the hanging wall of the faults in the late Paleozoic (**Error! Reference source not found.**d and Smelror et al., 2009 their profile C–D pp. 53))." lines 650–654. Also added

reference to Hassaan et al. (2021) and Hassaan (2021) lines 53. Added ", and the southeastern Norwegian Barents Sea (Hassaan et al., 2021)" lines 570–571. Added "in the southeastern Norwegian Barents Sea (Hassaan et al., 2020a, 2020b, 2021; Hassaan, 2021)," lines 736–737.

Comment 8: none recommended by the referee's comment.

Comment 9: none.

Comment 10: see response to comment 7.

Comment 11: none recommended by the referee's comment.

Comment 12: added " Caledonian folding of Timanian thrusts also explains the weaker magnetic and gravimetric signal of Timanian faults at the location of major Caledonian synclines where Timanian faults were transported downwards and, therefore, may not show well on potential field data (e.g., major two, NE–SW- to N–S-trending, negative gravimetric anomalies in the Russian Barents Sea just west of Novaya Zemlya; **Error! Reference source not found.**a)." lines 864–868.

Comment 13: none recommended by the referee's comment.

Comment 14: none. To be addressed in a new manuscript focusing in Svalbard that is the natural progression of this work.

Comment 15: none. See supplementary figures and note the dataset used is publicly available.

Comment 16: none. See supplementary figures and note the dataset used is publicly available.

Comment 17: none. See supplementary figures and note the dataset used is publicly available.

Comment 18: none. See figure 3d.

Comment 19: none recommended by the referee's comment.

Comment 20: added "Magnetic and gravimetric anomalies not related to Timanian and Caledonian grains will not be discussed in the present study." lines 247–248.

Comment 21: none recommended by the referee's comment.

Comment 22: added "The boundary between Precambrian, lower Paleozoic and upper Paleozoic successions offshore are interpreted as major unconformities that truncate underlying reflections and fold structures (e.g., Figure 3a, b and c). The boundaries between Devonian–Permian and Mesozoic successions were tied to the Raddedalen-1, Plurdalen-1, and Hopen-2 exploration wells for offshore parts of the study area. The boundary between Devonian–Mississippian and Pennsylvanian–Permian onshore Svalbard are interpreted as a major unconformity truncating Devonian–Mississippian dykes (see Figure 3e)." lines 235–241.

Comment 23: added "This further explains why Timanian faults were not reactivated exclusively as strike-slip faults despite being oriented sub-orthogonal (c. 70°) to E–W Caledonian contraction. Portions of Timanian faults near the Caledonian collision zone were locally deformed into subvertical geometries suitable to accommodate lateral movement, whereas their counterparts retaining their moderate–low-angle dip away from the paleo-plate boundary were more prone to accommodate vertical movements. Moreover, lateral transport of rocks from the Caledonian collision front towards the inner portions of the Barents Sea in the east was hampered by rock units constituting the crust of the Barents Sea, northern Norway, northwestern Russia and other adjacent areas. Hence, Caledonian E–W contraction produced more easily N–S-trending folds (e.g., Figure 3b and e and Figure 4f), which extended almost all the way to Novaya Zemlya (Figure 5), whereas partial strike-slip reactivation was restricted to areas proximal to the Caledonian collision front (e.g., western Spitsbergen; Majka et al., 2008; Mazur et al., 2009; Faehnrich et al., 2020; Ziemniak et al., 2020)." lines 933–945.

Comment 24: none recommended by the referee's comment.

**Attached figures**

[Figure]

Figure 1: Gravimetric anomaly map over the Svalbard Archipelago from the Geological Survey of Norway (Skilbrei et al., 2000) showing a major negative anomaly in western Spitsbergen. The anomaly correlates with the early Cenozoic Central Tertiary Basin in central and southern Spitsbergen. However, since there are no lower Cenozoic sedimentary deposits north of Kongsfjorden (Kg) in northwestern Spitsbergen, it is very likely that the negative anomaly also

reflects basement attitudes, most likely a N–S- to NNW–SSE-trending synform. This synform is abruptly narrows across Kongsfjorden, thus suggesting fault offset. This offset is interpreted as being accommodated by the continuation of the Kongsfjorden–Cowanodden fault zone (KCFZ) in Kongsfjorden. The fault accommodated dominantly top-SSW reverse movements, which may very well explain the observed offset of the synform across the fjord. The narrowing of the gravimetric anomaly therefore most likely reflects uplift and partial erosion of the N–S-trending synform in northwestern Spitsbergen, which is supported by the absence of lower Cenozoic sedimentary rocks in the north. This interpretation shows that some of the Timanian faults presented in the present manuscript do extend west of Svalbard.